# SafeAuto: Knowledge-Enhanced Safe Autonomous Driving with Multimodal Foundation Models

## Abstract

Traditional autonomous driving systems often struggle to harmonize high-level reasoning with low-level control, leading to suboptimal and even unsafe driving behaviors. The emergence of multimodal large language models (MLLMs), capable of processing visual and textual data, presents an opportunity to unify perception and reasoning tasks within a single framework. However, integrating precise safety knowledge into MLLMs for safe autonomous driving remains a significant challenge. To address this, we propose SafeAuto, a novel framework that enhances MLLM-based autonomous driving systems by incorporating both unstructured and structured knowledge. In particular, we first propose the Place-Dependent Cross-Entropy (PDCE) loss function, which is specifically designed to enhance the accuracy of low-level control signal predictions when treating numerical values as text. To explicitly integrate precise safety knowledge into the MLLM to enable safe autonomous driving, we build a reasoning component for SafeAuto, which first parses driving safety regulations into first-order logic rules (e.g., "red light $\implies$ stop") and then integrates these rules into a probabilistic graphical model, such as a Markov Logic Network (MLN). The environment attributes, identified by attribute recognition models (e.g., detecting a red light), are used to form the predicates in MLN. In addition, the environmental attributes utilized for reasoning are also considered factors in retrieval to construct a Multimodal Retrieval-Augmented Generation (RAG) model, which aims to learn from past similar driving experiences more effectively. Extensive experiments demonstrate that SafeAuto significantly outperforms baselines across multiple datasets. By bridging the gap between high-level reasoning and low-level control, SafeAuto paves the way for more accurate, reliable, and safer autonomous driving, facilitating systems that learn effectively from experience, adhere to traffic regulations, and execute precise control actions.

## 1 Introduction

Autonomous Driving (AD) systems (Kim et al., 2018; Jin et al., 2023; Hu et al., 2023) have made significant strides in recent years, yet they often rely on separate modules for high-level decision-making (e.g., "the car should slow to a stop") and low-level control signal prediction (e.g., providing the specific speed or steering angle for the next few moments). However, these two aspects are inherently correlated, as high-level actions directly guide low-level control signals. This modular design often overlooks this correlation, leading to inefficiencies and less cohesive driving behaviors.

Recent advancements in *Multimodal Large Language Models* (MLLMs) (Liu et al., 2023b;a; Lin et al., 2023) offer a promising avenue to bridge the gap between high-level reasoning and low-level control in AD. These models provide a unified framework capable of processing and reasoning over multiple data modalities, such as images, videos, and text. Some recent works (Wang et al., 2023; Xu et al., 2024; Wang et al., 2024) have begun to leverage MLLMs to generate both high-level action descriptions and low-level control signals in an end-to-end manner. However, these approaches are predominantly data-driven and often fail to perform at human levels due to several limitations.

Firstly, for low-level action prediction, current approaches in adapting MLLMs generally follow two fashions. The first fashion treats the prediction of float numbers as text generation (Gruver et al., 2024; Xu et al., 2024), directly training the MLLM using cross-entropy (CE) loss for token

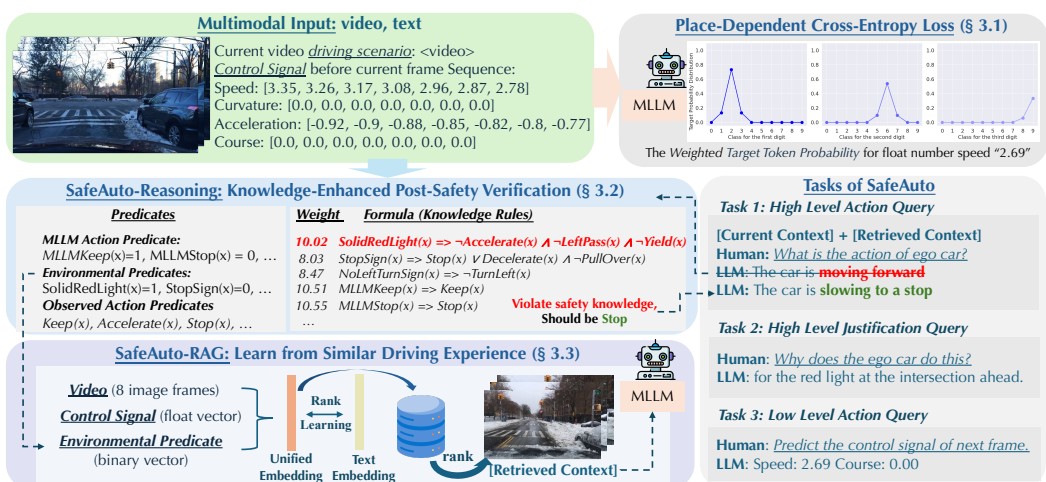

Figure 1: Overview of our **SafeAuto** pipeline for end-to-end high-level and low-level prediction in autonomous driving, featuring: (1) the Place-Dependent Cross-Entropy Loss (Section 3.1) for improved low-level numerical predictions using soft, weighted digit probability distributions; (2) Knowledge-Enhanced Post-Safety Verification (Section 3.2) with Markov Logic Networks to verify high-level actions against traffic rules; and (3) a Multimodal RAG (Section 3.3) training method that incorporates similar driving experiences via text-based rankings for better context-aware decision-making.

prediction. Some variations (Brohan et al., 2023; Sima et al., 2023) of this method involve tokenizing the prediction range into several bins and adding new tokens for each bin into the LLM's vocabulary, allowing the model to predict the corresponding bin token ID. However, these methods remain somewhat coarse compared to traditional regression techniques (Hu et al., 2023) using Mean Squared Error (MSE) loss. Alternatively, another fashion (Jin et al., 2024) employs a linear layer to decode the float number from the output hidden embeddings of the MLLM, enabling the use of MSE loss to train the model. While this approach may improve numerical accuracy, it compromises the autoregressive capability of the LLM, as the model can then only be purely used for numerical prediction and cannot perform any further QA-for example, handling high-level question-answering.

Additionally, regarding high-level action prediction, a significant limitation of current methods is their inability to effectively utilize both structured and unstructured knowledge when making decisions. Specifically, existing approaches often focus solely on data-driven techniques, inadequately incorporating structured knowledge such as traffic rules and safety constraints. Although some methods (Sima et al., 2023; Mao et al., 2023; Wang et al., 2024) attempt to include traffic regulations by embedding them into the model's context, this implicit approach is insufficient. Due to the inherent tendency of MLLMs to hallucinate, they may still generate unsafe or illegal actions. Meanwhile, while RAG (Lewis et al., 2020) has been employed in language models (Semnani et al., 2023; Zhang et al., 2024) to mitigate issues like hallucination by incorporating relevant information from external sources, few works (Yuan et al., 2024) have fully exploited and combined the rich multimodal data inherent in autonomous driving contexts—such as videos, images, and control signals—to learn from past driving experiences as unstructured knowledge.

To address these challenges, we propose a novel framework **SafeAuto** that enhances MLLMs for autonomous driving through three key contributions as shown in Figure 1: (1) **Place-Dependent Cross-Entropy (PDCE) Loss:** We propose a PDCE loss that retains the autoregressive nature of the MLLM while behaving like an MSE loss during training. This loss function improves numerical prediction accuracy without compromising the model's language generation abilities. (2) **Knowledge-Enhanced Post-Safety Verification:** We employ *Markov Logic Networks* (MLNs) (Richardson & Domingos, 2006) to explicitly encode domain knowledge and structured traffic rules into the decision-making process of the MLLM. This knowledge-enabled reasoning allows us to verify and correct the high-level actions suggested by the MLLM, ensuring they comply with traffic regulations and safety constraints. (3) **Multimodal RAG for Autonomous Driving:** We introduce a method that utilizes video data, control signals, and the environmental predicates used in the MLN to retrieve similar driving experiences. By learning a joint embedding across these modalities based on the ranking derived from text description of the current scenario—which contain rich semantic information—we can effectively leverage past experiences to inform current decision-making.

By integrating these components, **SafeAuto** provides a comprehensive solution to the challenges faced by current MLLMs in autonomous driving. We evaluate our approach on two benchmark

datasets: *BDD-X* (Kim et al., 2018) and *DriveLM* (Sima et al., 2023), both featuring low-level control signals and high-level action descriptions. Our experimental results demonstrate significant improvements in both low-level control accuracy and high-level action prediction. First, for low-level prediction on the BDD-X dataset, it reduces the Root Mean Square Error (RMSE) for speed and course predictions from the state-of-the-art (SOTA) values of $0.69$ and $4.48$ to $0.65$ and $3.85$, respectively. Furthermore, on the DriveLM dataset, it decreases the Average Displacement Error (ADE) for motion prediction from 1.51 to 0.84. Second, for high-level prediction on the BDD-X dataset, our method boosts the high-level action from SOTA of 260.8 to 337.4 under metric CIDEr, while on the DriveLM dataset, the high-level behavior prediction accuracy is improved from the SOTA value of 61.60% to 74.60%.

## 2 RELATED WORK

Advancements in autonomous driving have produced comprehensive frameworks like UniAD (Hu et al., 2023), which integrates modules for tracking, mapping, motion prediction, and occupancy estimation for low-level planning. However, UniAD lacks high-level action descriptions and textual justifications. To address high-level explanations, Kim et al. (2018) proposed an attention-based video-to-text model generating explanations of current driving actions. Similarly, ADAPT (Jin et al., 2023) employs a video Swin Transformer (Liu et al., 2022) to extract video tokens for separate high-level and low-level action predictions.

The emergence of MLLMs enables unified end-to-end generation of both high-level and low-level outputs. Most of these works often treat numerical control signals as text, training models using token prediction with cross-entropy loss. For example, DriveGPT4 (Xu et al., 2024) just treats low-level control signals as text, fine-tuning an MLLM to sequentially predict high-level and low-level actions in a conversational manner using the BDD-X dataset. DriveLM-Agent (Sima et al., 2023), influenced by RT-2 (Brohan et al., 2023), discretizes waypoints into bins, expanding the tokenizer vocabulary accordingly and fine-tuning the BLIP-2 (Li et al., 2023). While this facilitates end-to-end training, it remains coarse compared to UniAD (Hu et al., 2023), which uses MSE loss. Time-LLM (Jin et al., 2024) decodes numerical predictions directly from output embeddings using a linear layer with MSE loss but diminishes the language model's autoregressive capabilities, limiting high-level question-answering abilities. Additionally, Tan et al. (2024) suggest that employing the LLM backbone in this way does not enhance regression performance. In contrast, we propose a novel PDCE loss that adapts the cross-entropy loss for numerical training to behave more like MSE loss while preserving the model's ability to perform high-level question-answering.

Further advancements involve integrating perception and planning tools into the MLLM context. Agent-Driver (Mao et al., 2023) incorporates modules from UniAD into an MLLM framework, serving as a language agent for autonomous driving. OmniDrive (Wang et al., 2024) introduces a framework combining 3D perception, reasoning, and planning. However, these methods remain purely data-driven and lack explicit safety verification for generated actions. Given the safety-critical nature of autonomous driving, ensuring that output actions are safe and compliant with traffic rules is essential. To address this, we incorporate extracted knowledge—specifically structured traffic rules—into a probabilistic graphical model like a Markov Logic Network (MLN) for explicit post-safety verification. Besides, RAGDriver (Yuan et al., 2024) further enhances reasoning by retrieving similar driving experiences through triplet loss-based metric learning. We extend this approach by developing a more flexible and efficient retrieval system, directly training a joint embedding based on multimodal inputs to learn relative rankings from text similarity. Most importantly, we find that the incorporation of binary structured environmental predicates (e.g., the presence of a stop sign) from the previous reasoning components, namely MLNs, significantly improves retrieval performance.

## 3 SAFEAUTO

**Motivation.** Recent studies have begun to explore the integration of MLLMs into autonomous driving systems to enhance both high-level reasoning and low-level control actions. As illustrated in Figure 1, the MLLM receives a sequence of current driving images or videos, accompanied by textual descriptions of historical control signals, including speed, curvature, acceleration, and course, as inputs. Then, during the conversation, the model is expected to answer three types of queries: (1) *High-Level Action Queries*: These queries request a textual description of the action that the current ego vehicle is performing or should perform. For example, when asked *"What is the action of the ego car?"*, the MLLM is expected to respond with an answer like *"The car is slowing down to stop"*. (2) *High-Level Justification Queries*: These queries seek an explanation for the action provided by the MLLM. For instance, *"Why is the ego car doing this?"* prompts the model to justify the action,

such as *"Because there is a red light at the upcoming intersection"*. (3) *Low-Level Action Queries*: These queries request specific control signals or trajectories that the vehicle should execute in the future. For example, the query *"Predict the control signals for the next frame"* would elicit a response like *"Speed: 2.69, Course: 0.00"*, which can then be translated into actual control commands for the autonomous vehicle. Typically, low-level action queries follow high-level action and justification queries, ensuring that generated control signals are conditioned on prior high-level actions for more accurate and coherent driving control.

**Overview.** In this section, we detail the three main components proposed within this framework, each elaborated in subsequent sections: (1) a Place-Dependent Cross-Entropy Loss function for improved low-level action prediction (Section 3.1); (2) Knowledge-Enhanced Post-Safety Verification using Markov Logic Network (MLN) for high-level action prediction (Section 3.2); (3) Multimodal Retrieval-Augmented Generation (RAG) for learning from similar driving experiences (Section 3.3). In summary, during training, we first fine-tune the underlying MLLM using the PDCE loss with the retrieved context to enhance the accuracy of low-level action predictions. During evaluation, we retrieve the top $K$ similar driving experiences from the training database, generate high-level actions using the MLLM, and apply post-safety verification using the MLN to ensure that the actions comply with traffic rules and safety constraints.

## 3.1 PLACE-DEPENDENT CE LOSS

In existing approaches that utilize MLLMs for autonomous driving, the next-token prediction loss-specifically, the cross-entropy loss is commonly applied uniformly across all prediction tasks, including numerical value predictions. However, for numerical regression tasks, it is standard practice to use the Mean Squared Error (MSE) loss, as it directly penalizes the squared difference between the predicted and true values. A fundamental difference between CE loss and MSE loss lies in how they handle proximity to the target: MSE loss decreases as the prediction gets numerically closer to the target value, whereas CE loss does not necessarily exhibit this property.

This issue is also empirically observed in the speed prediction distribution when using the original CE loss to fine-tune the MLLM on the BDD-X dataset, as shown in Figure 3 (a), which displays predictions over 200 samples given the same input driving context with temperature as 1.0. As we can see, it reveals two distinct peaks, indicating that predictions closer to the ground truth value of "12.46" do not necessarily occur with higher frequency or lower loss, contrary to the behavior expected from MSE loss. A natural solution might be to append a MLP to the MLLM to decode the output hidden embeddings into corresponding float values and thus use MSE loss for fine-tuning. However, currently, incorporating an MLP in this manner usually disrupts the autoregressive token generation capability of the MLLM, rendering it unable to perform high-level action queries or engage in continued conversation. Essentially, the model becomes a pure transformer encoder (Tan et al., 2024) used solely for regression tasks, losing its language generation functionalities critical for interactive and interpretative tasks.

**PDCE loss.** To overcome these challenges, we adapt the CE loss to function more like MSE loss while maintaining textual predictions. Consider the previous example of predicting the float number "12.46." Originally, the MLLM is trained to maximize the probabilities $p(`1') \cdot p(`2' \mid `1') \cdot p(`.' \mid `12') \cdot p(`4' \mid `12.') \cdot p(`6' \mid `12.4')$ by minimizing the CE loss with one-hot labels. However, as we see before, this does not ensure that predictions closer to the target value—such as "11.99" have a lower loss compared to more distant predictions like "2.46," because each digit's probability is treated separately and with equal importance (with all weights set to one).

To make the CE loss behave more like MSE loss, we make two modifications: (1) *Digit-Level Loss Adjustment*: Instead of using one-hot hard target labels for each digit, we employ a soft target discrete distribution $\mathcal{D}(\mu, \sigma)$ centered around the target digit $\mu$, which assigns higher probabilities to digits closer to the target, allowing the loss to reflect numerical proximity. Specifically, we leverage a Gaussian distribution $\mathcal{G}(\mu, \sigma)$ to construct $\mathcal{D}(\mu, \sigma)$ for each digit (ranging from 0 to 9), while other distribution methods are also workable. We then compute the loss for each digit as the KL divergence between the target distribution $\mathcal{D}(\mu, \sigma)$ and the predicted probability distribution $\mathcal{P}$ on all digits output by the MLLM. (2) *Place-Level Weighting*: Instead of treating all digits equally important, we apply decreasing weights from the first-place digit to the last-place digit based on cumulative probabilities. For example, for float number "12.46", the weight for the loss on digit '2' is the probability of '1' under $\mathcal{D}(1, \sigma)$, and the weight for digit '4' is the cumulative probability of '1' multiplied by the probability of '2' under $\mathcal{D}(2, \sigma)$. In this way, errors in more significant digits have a greater impact on the loss, while other weighting designs can also be explored.

```
# str_num: a string representing a float number (excluding
the decimal point '.') with N digits
# logits: the logits distribution output from MLLM for each
digit in str_num, with a shape of N * 10
# sigma: the standard deviation of the Gaussian distribution

# Precompute the digit-level probability distributions
from scipy.stats import norm
distribution_dict = {}
for num in range(10):
  prob_distribution = np.array([norm(num, sigma).cdf(i + 0.5)
    - norm(num, sigma).cdf(i - 0.5) for i in range(10)])
  prob_distribution /= prob_distribution.sum()
  distribution_dict[str(num)] = prob_distribution

# Calculate weights for each digit position
tgt_probs = []
Weight = 1.0
for digit in str_num:
  # The place-level weighting
  digit_probs = distribution_dict[digit] * weight
  weight *= digit_probs[int(digit)]
  tgt_probs.append(digit_probs)

tgt_probs = np.array(tgt_probs)
# Compute the KL loss, constants are ignored
loss = - (tgt_probs * log_softmax(logits, axis=1)).sum()
```

Figure 2: Numpy-like pseudocode for the core implementation of PDCE loss.

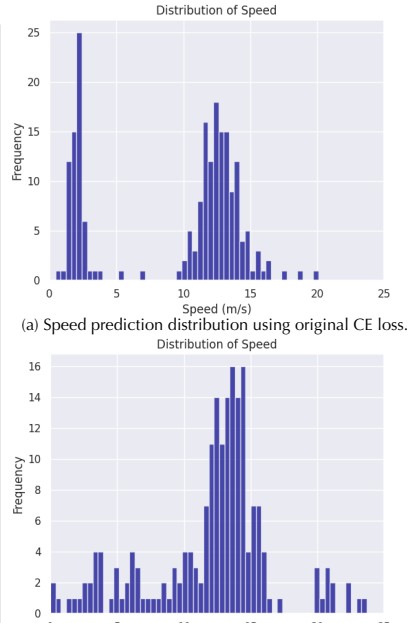

(a) Speed prediction distribution using original CE loss.

(b) Speed prediction distribution using PDCE loss.

Figure 3: Sampled speed prediction distribution under different losses.

As a result, the final loss is the weighted sum of the KL divergence between the probabilities generated by the MLLM and the target digit-level soft probability distributions. Mathematically, it can be expressed as: $\sum_{i=1}^{n} w_i \cdot \text{KL}(\mathcal{P}_i \parallel \mathcal{D}(\mu_i, \sigma))$, where $n$ is the number of digits, $\mu_i$ is the $i$-th digit, $\mathcal{P}_i$ represents the probability distribution over the possible digits for the $i$-th digit position from MLLM, $w_i$ represents the weight based on the iterative probability calculation of previous digits for the $i$-th digit, the pseudo-code for implementing this loss during practice is provided in Figure 2. Notice that when $\sigma$ is set to 0, the loss reduces to the original definition of joint CE loss for the entire numeric string. The new prediction distribution using the new loss with $\sigma = 0.35$ is demonstrated in Figure 3 (b). As shown, the distribution exhibits higher frequencies for predictions closer to the ground truth, aligning with the desired outcome and verifying the intuition behind our method.

### 3.2 KNOWLEDGE-ENHANCED POST-SAFETY VERIFICATION WITH SAFEAUTO-REASONING

Currently, most methods for autonomous driving that utilize MLLMs are still purely data-driven. While these data-driven approaches have led to significant advancements, they may not be entirely suitable for safety-critical scenarios like autonomous driving, where reliability and strict adherence to safety regulations are paramount. To address this concern, we propose incorporating Probabilistic Graphical Models (PGMs) to verify the safety of the high-level actions suggested by the underlying MLLM. Specifically, in this paper, we focus on demonstrating how to adopt Markov Logic Networks to integrate domain knowledge and traffic rules into the decision-making process, while other variants are also applicable. In this section, we begin by explaining what MLNs are and how they apply to our autonomous driving context.

**Definition.** Essentially, an MLN consists of a set of first-order logic formulas, each associated with a weight that reflects the strength or confidence of that formula. These weights allow us to model uncertainty and handle exceptions in real-world knowledge. In our autonomous driving scenario, we use MLNs to model traffic rules and safety constraints. For example, a traffic rule like *"If there is a stop sign, then the vehicle should stop or decelerate"* can be represented as the logical formula: `StopSign(x)` $\implies$ `Stop(x)` $\lor$ `Decelerate(x)`, where x represents the current driving scenario. Here, predicates such as `StopSign(x)`, `Stop(x)`, and `Decelerate(x)` are logical functions that return true or false, indicating whether the condition holds in scenario $x$.

Formally, in MLNs, *predicates* are logical functions defined over a set of constants $\mathcal{V} = \{v_1, v_2, \ldots, v_N\}$, where each $v_i$ represents an object or concept in the domain, such as "stop sign" or "red light." A predicate takes these constants as arguments and returns a truth value: $k(\cdot) : \mathcal{V} \times \cdots \times \mathcal{V} \to 0, 1$. While *formulas* are logical statements composed of predicates and logical connectives (e.g., $\implies$, $\land$, $\lor$), with each formula $f$ associated with a weight $w_f$ indicating its importance. Then, an MLN defines a joint probability distribution over all possible assignments of truth values to the ground predicates (predicates with specific constants assigned). The prob-

ability of a particular world (an assignment of truth values to all ground predicates) is given by: $P(\mathbf{X}) = \frac{1}{Z} \exp\left(\sum_{f \in \mathcal{F}} w_f \sum_{a_f \in \mathcal{A}_f} \phi_f(a_f)\right)$, where $\mathbf{X}$ is the set of all ground predicates, $\mathcal{F}$ is the set for all formulas $f$, $Z$ is the partition function ensuring the distribution sums to one, $\phi_f(a_f)$ is the potential function for formula $f$ with assignment $a_f$ (which equals 1 if $f$ is true under $a_f$ and 0 otherwise), and $\mathcal{A}_f$ is the set of all possible assignments to the arguments of formula $f$.

**Autonomous Driving Context.** In our case, we categorize predicates into *unobserved predicates* ($\mathcal{U}$) and *observed predicates* ($\mathcal{O}$). Specifically, the unobserved predicates $\mathcal{U}$, are the *Main Action Predicates* that encompass potential actions a vehicle might or should take, such as `Accelerate(x)`, `Stop(x)`, and `TurnLeft(x)`, among others. While observed predicates $\mathcal{O}$ include: (1) *MLLM Action Predicates*: This set also includes the number of predicates for the main action such as `MLLMAccelerate(x)`, `MLLMStop(x)`, and `MLLMTurnLeft(x)`, which indicate the high-level actions suggested by the MLLM. Generally, we would prompt GPT4o to map the high-level action descriptions generated by the MLLM to the corresponding truth values of each predicate. Then, we introduce formulas like `MLLMAccelerate(x)` $\Rightarrow$ `Accelerate(x)` to reflect the influence of the original high-level action decision made by the MLLM. (2) *Environmental Predicates*: These predicates describe the surrounding environment. For example, `StopSign(x)` indicates whether there is a stop sign in the current driving scenario, and `SolidRedLight(x)` indicates whether the traffic light ahead is red. The truth values of these predicates can be extracted from video data using any object detector. These object predicates are combined with the main action predicates to form logical formulas based on traffic rules extracted from the California Driver Handbook [1]. Specifically, we first crawled the handbook and used GPT4o to map the rules into corresponding first-order logical formulas, e.g., `StopSign(x)` $\implies$ `Stop(x)` $\lor$ `Decelerate(x)` $\land$ `¬PullOver(x)`, details are deferred to Appendix A. In addition to object-related predicates, we define predicates associated with historical control signals. For instance, the predicate `HCSTurnLeft(x)` determines whether the ego vehicle had recently turned left, based on historical control signals. These predicates are integrated with main action predicates to effectively reflect the vehicle's inherent tendencies in its actions.

**Inference.** Our goal is to infer the most probable assignment of the unobserved main action predicates $\mathcal{U}$ given the observed predicates $\mathcal{O}$. To determine the safest and most appropriate action, we perform inference by maximizing the conditional probability $P(\mathcal{U}|\mathcal{O})$. Specifically, we seek the assignment to the main action predicates $\mathcal{U}$ that maximizes this probability $\mathcal{U}^* = \arg\max_{\mathcal{U}} P(\mathcal{U}|\mathcal{O})$. Since the possible worlds for $\mathcal{U}$ (i.e., the possible assignments to the main action predicates) are inherently limited—a vehicle cannot simultaneously accelerate and decelerate or turn left and right—the inference process is thus computationally efficient. The detailed specifics of the possible worlds can be found in Appendix A.6.

**Training.** The training of the MLN is straightforward and involves learning the weights $w_f$ of the formulas to maximize $P(\mathcal{U}|\mathcal{O})$. In our approach, we utilize a mix of real and simulated data for training. The real data serves as the ground training data, provided by datasets such as BDD-X, while the simulated data allows us to model various driving conditions. This includes rare or dangerous scenarios not present in the real data, by simulating different truth values for the predicates to perform inference. Details are deferred to Appendix A.4.

**Safety Verification.** Initially, we collect observed grounded environmental predicates and the MLLM action predicates from high-level actions generated by the MLLM, extracted through object detector and prompting with GPT4o. These predicates undergo inference within the trained MLN. If the MLN's final main action predicate output contradicts the MLLM's suggested action—suggesting a potential safety violation or a breach of critical traffic rules, we overwrite the original high-level action query based on the MLN's output and re-prompt the MLLM to generate a new high-level action, as depicted in Figure 1. Further details are available in Appendix A.5.

In this way, the MLN serves as a post-verification layer that can override unsafe suggestions from the MLLM, enhancing the overall reliability of the autonomous driving system.

### 3.3 SAFEAUTO-MULTIMODAL RETRIEVAL-AUGMENTED GENERATION

In this section, we introduce a novel training method for constructing a unified embedding that effectively integrates multiple modalities—current driving videos, historical control signals, and observed environmental predicate information from Section 3.2. Specifically, we aim to train the joint embedding to mirror the similarity rankings derived from the embedding of the textual descriptions

---

[1] https://www.dmv.ca.gov/portal/handbook/california-driver-handbook/

for the current driving scenarios, which encapsulate the semantic information of all modalities during training. This approach facilitates the retrieval of similar driving experiences, enabling the ego vehicle to make more informed and context-aware decisions in current driving situations.

**Different Modality.** (1) *Image/ Video Embedding:* for the image or video modality, we utilize the pre-trained LanguageBind encoder (Zhu et al., 2024). This encoder processes an input image in $\mathbb{R}^{256 \times 1024}$, while processing video into eight frames and generates a video embedding in $\mathbb{R}^{2048 \times 1024}$. For simplicity and to reduce computational complexity, we apply global average pooling over the first dimension for both modalities here, resulting in a compressed embedding $\mathcal{Z}_v \in \mathbb{R}^{1 \times 1024}$ for use in subsequent experiments. (2) *Control Signal Vector:* the control signals are numerical values representing various aspects of the ego vehicle's historical state, such as speed, curvature, acceleration, and course. In datasets like BDD-X, each of these four types of control signals contains seven historical values (excluding the current frame), resulting in a total of $N = 4 \times 7 = 28$ values. We concatenate these values into a single vector $\mathcal{Z}_c \in \mathbb{R}^{1 \times N}$, which serves as the initial control signal vector. (3) *Environmental Predicate Vector:* These environmental predicates introduced in Section 3.2 are binary indicators of certain conditions or observations (e.g., presence of a stop sign, status of a traffic light). We encode these predicates into a single binary vector $\mathcal{Z}_p \in \{0, 1\}^{1 \times M}$, where $M$ is the number of the whole environmental predicates. Empirically, we found that including this explicit binary representation significantly boosts retrieval performance, as demonstrated in Section 5. This enhancement may be attributed to the reduction of noise inherent in the raw video embeddings or control signals; the binary predicates provide a clearer and more robust representation of essential environmental information.

**Unified Embedding Construction.** The central question is: *How can we train a unified embedding that effectively combines these different modalities for similarity computation and retrieval?* A key insight is that textual descriptions of the current driving scenario typically encompass all relevant semantic information, reflecting aspects of the video, control signals, and predicates. For instance, a text that concatenates action and justification—such as "*The car is slowing to a stop for the red light at the intersection ahead*" as shown in Figure 1 captures the essence of all three modalities. This comprehensive representation is particularly valuable for ranking the most similar driving scenarios. However, such ground text descriptions are often not available during evaluation. Building on this intuition, we propose learning a unified embedding that aligns these modalities in a shared space, akin to how text embeddings represent semantic information.

**Training.** Specifically, we first utilize individual projectors to map each input vector—$\mathcal{Z}_v$, $\mathcal{Z}_c$, and $\mathcal{Z}_p$—into aligned embeddings $\mathcal{Z}'_v$, $\mathcal{Z}'_c$, and $\mathcal{Z}'_p$, each with the same dimension and normalized to a unit $\ell_2$ norm. We then introduce weighting factors $w_v$, $w_c$, and $w_p$ to modulate the contribution of each modality in the input aligned embedding. The final unified embedding is then computed as $\mathcal{Z}_u = \texttt{Projector}(w_v \mathcal{Z}'_v + w_c \mathcal{Z}'_c + w_p \mathcal{Z}'_p) \in \mathbb{R}^{1 \times H}$. While other design choices are also feasible, we found through experimentation that this configuration provides better controllability. Let $Z_t \in \mathbb{R}^{1 \times I}$ represent the text embeddings of scenario descriptions, e.g., the concatenation of high-level actions and justifications. Our goal is for the unified embedding $\mathcal{Z}_u$ to mirror the relational properties of text embeddings derived from scenario descriptions, particularly in terms of similarity rankings. Then, during training, we will first randomly sample a batch of cases with unified embeddings $\mathcal{Z}'_u \in \mathbb{R}^{B \times H}$ and the corresponding text embeddings $Z'_t \in \mathbb{R}^{B \times I}$ with batch size $B$. We then minimize the KL divergence between the inter-similarity distributions derived from the unified embeddings $\mathcal{Z}'_u$ and those from the text embeddings $Z'_t$. In specific, we compute the similarity matrices (assuming each row in both $\mathcal{Z}'_u$ and $Z'_t$ have been normalized to unit $\ell_2$ norm) as follows: $S_u = \mathcal{Z}'_u (\mathcal{Z}'_u)^\top$ and $S_t = Z'_t (Z'_t)^\top$. Then, the loss function aims to minimize the mean of the divergence between the logits $S'_u$ and the target logits $S'_t / \tau$ across each row. Here, $\tau$ is a temperature parameter that adjusts the sharpness of the target probability distributions. A lower $\tau$ focuses learning on the most similar (positive) examples, crucial for retrieval tasks where pinpointing the closest matches is essential. By aligning the similarity distributions, we ensure the unified embeddings preserve the relative rankings observed in text embeddings, enabling effective retrieval without relying on the unavailable ground textual descriptions during inference.

## 4 EXPERIMENTS

In this section, we present our experimental results on two datasets: the BDD-X dataset (Kim et al., 2018) and the DriveLM dataset (Sima et al., 2023), both of which contain high-level action questions and low-level control questions. Specifically, we find that: (1) when using the Place-Dependent

Table 1: High-level action and justification evaluation on BDD-X dataset. B4, C, and M represent BLEU4, CIDEr, and METEOR, respectively.

| Method | Action | | | Justification | | |
|---|---|---|---|---|---|---|
| | B4 ↑ | C ↑ | M ↑ | B4 ↑ | C ↑ | M ↑ |
| ADAPT | 34.6 | 247.5 | 30.6 | **11.4** | 102.6 | **15.2** |
| DriveGPT4 | 30.0 | 214.0 | 29.8 | 9.4 | 102.7 | 14.6 |
| RAGDriver | 34.3 | 260.8 | 30.7 | 11.1 | **109.1** | 14.8 |
| SafeAuto | **38.6** | **337.4** | **35.5** | 9.4 | 96.0 | 14.0 |

Table 2: High-level behavior and low-level motion prediction evaluation on DriveLM dataset.

| Method | High-Level Behavior | | | Motion |
|---|---|---|---|---|
| | Acc ↑ | Speed ↑ | Steer ↑ | ADE ↓ |
| UniAD-Single | - | - | - | 1.80 |
| UniAD-Full | - | - | - | **0.80** |
| BLIP-RT-2 | - | - | - | 2.63 |
| DriveLM-Agent | 61.60 | 65.40 | 81.61 | 1.51 |
| SafeAuto | **74.60** | **81.61** | **81.90** | 0.84 |

Table 3: Low-level control signal prediction evaluation on BDD-X dataset.

| Method | Speed | | | | | | Course | | | | | |
|---|---|---|---|---|---|---|---|---|---|---|---|---|
| | RMSE ↓ | $A_{0.1}$ ↑ | $A_{0.5}$ ↑ | $A_{1.0}$ ↑ | $A_{5.0}$ ↑ | $A_{10.0}$ ↑ | RMSE ↓ | $A_{0.1}$ ↑ | $A_{0.5}$ ↑ | $A_{1.0}$ ↑ | $A_{5.0}$ ↑ | $A_{10.0}$ ↑ |
| ADAPT | 2.68 | 11.77 | 31.79 | 47.48 | 92.75 | 95.87 | 5.87 | 54.49 | 86.39 | 91.06 | 97.36 | 98.20 |
| DriveGPT4 | 1.09 | **56.93** | 77.77 | 87.97 | 99.00 | 99.57 | 4.57 | 69.22 | 79.14 | 84.47 | 95.72 | 96.74 |
| RAGDriver* | 0.69 | 51.12 | 85.54 | 94.49 | **99.81** | **99.91** | 4.48 | 74.32 | 88.69 | 93.12 | **98.30** | 99.10 |
| SafeAuto | **0.65** | 55.49 | **88.84** | **95.34** | **99.81** | **99.91** | **3.85** | **76.26** | **89.68** | **94.11** | **98.30** | **99.25** |

* Notice, RAGDriver leveraged the test data for training the retriever.

Cross-Entropy loss, the numerical prediction of float numbers is significantly improved; (2) with the post-safety knowledge-enhanced verification via MLN, many dangerous high-level actions have been corrected; (3) the incorporation of Multimodal RAG, specifically integrating environmental predicate information from the MLN component, leads to significant improvements in the MLLM's high-level prediction performance. Notably, our framework is *plug-and-play* and can be directly applied to any new methods based on MLLMs. All experiments are conducted on eight NVIDIA A6000 GPUs.

**Datasets and Tasks.** (a) *BDD-X:* In this work, we adopt the processed version from RAG-Driver (Yuan et al., 2024), where the task involves using an input video along with control signals from the past seven frames as context for a conversation that focuses on three types of questions: (i) high-level action queries, (ii) high-level justification queries, and (iii) low-level action predictions for speed and course in the next frame. This processed dataset contains 16,390 training video QA conversations and 2,123 test conversations. (b) *DriveLM:* The DriveLM dataset is built upon the nuScenes dataset (Caesar et al., 2020). In this work, we primarily focus on tasks that involve using six multi-view images from the current frame, control signals, and trajectory positions from the past three seconds as input context. The conversation concentrates on: (i) planning for possible high-level safe actions, (ii) high-level behavior involving predicting speed and steering actions, which serve as multiple-choice questions, and (iii) low-level motion, predicting 2D trajectories for the next three seconds, similar to UniAD (Hu et al., 2023). We filter instances to include only those with a prediction horizon of at least 3 seconds, resulting in a final dataset of 3,447 training conversations and 685 test conversations.

**Model.** We use the pretrained Video-LLaVA (Lin et al., 2023) with Vicuna 1.5 7B (Zheng et al., 2023) as the base LLM for fine-tuning. We fine-tune the model for 2 epochs with a batch size of 128 on the BDD-X dataset and for 4 epochs with a batch size of 64 on the DriveLM dataset, using a learning rate of $5 \times 10^{-2}$.

**Experimental Details.** (a) *PDCE loss:* During the fine-tuning of the MLLM, we initialize $\sigma$ in $\mathcal{D}(\mu, \sigma)$ at a small value of 0.01 and geometrically increase it after each optimization step until it reaches the predefined value of $\sigma = 0.35$. This gradual increase helps stabilize the training process. Besides, to balance the loss among various float numbers, we standardize their representation by using consistent digit lengths in text form. For instance, on the BDD-X dataset, each number is formatted to five digits, such as representing 8.1 as "08.100" during training, whereas for the DriveLM dataset, we use a four-digit format. (b) *Post-safety verification via MLN:* we fine-tune YOLOv8 (Jocher et al., 2023) as the object detector for both traffic lights and signs. For the BDD-X dataset, we define 16 action predicates, 20 environmental predicates, and 35 formulas based on traffic rules. Similarly, for the DriveLM dataset, we define 7 action predicates, 29 environmental predicates, and 29 formulas. Among these, 10 environmental predicates are specifically derived from the nuScenes map expansion and pertain to lane markings. Further details are provided in Appendix A. (c) *Multimodal RAG:* we consistently employ four-layer multilayer perceptrons (MLPs) as projectors to obtain aligned embeddings for each modality and to generate the final unified embedding, and we use sentence-t5-xl (Ni et al., 2022) as our text encoder. The weighting factors $w_v$ and $w_c$ are both set to 0.4, while the weight for the predicate embedding $w_p$ is set to 0.2. We consistently set the learning rate to 0.001 and the temperature parameter $\tau$ to 0.5. For the BDD-X

Table 4: Ablation study of the contribution from each module in SafeAuto focusing on high-level action and justification assessment on the BDD-X dataset. "Acc" denotes the high-level action predicates accuracy.

| Method | Action | | | | Justification | | |
|---|---|---|---|---|---|---|---|
| | B4 ↑ | C ↑ | M ↑ | Acc ↑ | B4 ↑ | C ↑ | M ↑ |
| Base | 30.8 | 221.5 | 29.2 | 61.75 | 7.8 | 85.4 | 13.2 |
| PDCE | 31.4 | 231.4 | 29.3 | 61.94 | 7.9 | 84.2 | 13.2 |
| PDCE + MLN | 31.5 | 232.2 | 29.4 | 62.97 | 7.9 | 84.5 | 13.2 |
| PDCE + RAG | 38.2 | 334.8 | 35.3 | 91.00 | **9.4** | 95.5 | 13.9 |
| PDCE + MLN + RAG | **38.6** | **337.4** | **35.5** | **92.18** | **9.4** | **96.0** | **14.0** |

Table 5: Ablation study of the contribution from each module in SafeAuto on both high-level and low-level predictions using the DriveLM dataset.

| Method | High-Level Behavior | | | Motion |
|---|---|---|---|---|
| | Acc ↑ | Speed ↑ | Steer ↑ | ADE ↓ |
| Base | 60.58 | 64.67 | 80.29 | 0.86 |
| PDCE | 63.21 | 67.88 | 79.27 | 0.85 |
| PDCE + MLN | 66.86 | 71.39 | 80.29 | 0.85 |
| PDCE + RAG | 74.01 | 79.27 | 81.61 | **0.84** |
| PDCE + MLN + RAG | **74.60** | **79.85** | **81.90** | **0.84** |

dataset, the model is trained for 100 epochs with a batch size of 2,048 and uses $K = 2$ retrieval examples. For the DriveLM dataset, the model is also trained for 100 epochs but with a batch size of 512 and uses $K = 1$ retrieval example.

**Baselines.** (a) On the *BDD-X* dataset, we compare our method with several baselines: (1) *ADAPT* (Jin et al., 2023), a state-of-the-art video transformer-based method that provides high-level and low-level answers using two separate branches; (2) *DriveGPT4* (Xu et al., 2024), the first work to provide both high-level action descriptions and low-level vehicle control signals in an end-to-end fashion using an MLLM; and (3) *RAGDriver* (Yuan et al., 2024), a state-of-the-art method that leverages triplet loss to train multimodal retrieval models for autonomous driving. (b) For the *DriveLM* dataset, we use: (1) *DriveLM-Agent*, the current state-of-the-art method that employs graph-based visual question answering to improve high-level responses and uses motion tokenization for low-level prediction; (2) *UniAD* (Hu et al., 2023), the state-of-the-art method on the nuScenes dataset used here for comparing low-level predictionswe consider two versions: UniAD (Full), which utilizes the entire historical video input, and UniAD (Single), a variant modified to use only the current frame's input for a fair comparison; and (3) *BLIP-RT-2*, which fine-tunes BLIP-2 (Li et al., 2023) on the DriveLM data and utilizes trajectory tokenization as proposed in RT-2 (Brohan et al., 2023).

**Metrics.** (a) For the *BDD-X* dataset, we adopt widely used metrics for high-level prediction, including 4-gram BLEU (B4) (Papineni et al., 2002), METEOR (M) (Banerjee & Lavie, 2005), and CIDEr (C) (Vedantam et al., 2015). For low-level prediction, we use the Root Mean Square Error (RMSE) for both steering angle (in degrees) and speed (in meters per second). We also present "tolerant accuracy" metrics, $A_\delta$, representing the accuracy of predictions when binarized as being within a tolerance threshold $\delta$ of the ground truth. (b) For the *DriveLM* dataset, the high-level behavior questions are multiple-choice problems concerning speed and steering. We report the overall accuracy, as well as individual accuracies for speed and steering predictions. For low-level trajectory prediction, we use the Average Displacement Error (ADE), as in UniAD, which indicates the average $\ell_2$ distance between the predicted trajectory and the ground truth trajectory and is calculated as the average of the errors at the 1st, 2nd, and 3rd seconds.

**Results.** (a) *BDD-X* Dataset: The final results for high-level prediction, including both action and justification, are presented in Table 1, while the low-level predictions for speed and course are shown in Table 3. For high-level action prediction, SafeAuto improves performance by 11.6%, 29.4%, and 15.6% for the BLEU4, CIDEr, and METEOR metrics, respectively. Although the justification performance is slightly lower than the state-of-the-art method, it still significantly outperforms the vanilla fine-tuned Video-LLaVA model, as demonstrated in Section 5. For low-level control signal prediction, SafeAuto achieves further reduction of 5.8% in RMSE for speed prediction and 14.1% in RMSE for course prediction. The contributions of each component to the overall performance are detailed in Section 5. (b) *DriveLM* Dataset: The final results are demonstrated in Table 2. For high-level behavior prediction, SafeAuto improves accuracy by 13.00% compared to the SOTA baseline DriveLM-Agent. For low-level motion prediction, it achieves a further reduction of 44.4% in ADE over the DriveLM-Agent. Notably, the ADE of SafeAuto is even comparable to UniAD (Full) which is trained purely for low-level prediction.

## 5 ABLATION STUDY

In this section, we conduct various ablation studies on our framework to investigate the impact of each module and different hyperparameters, as described in Section 3. For simplicity, we denote the base modeltrained directly on conversation data using Video-LLaVA without incorporating any of the modules introduced in our paperas 'Base'.

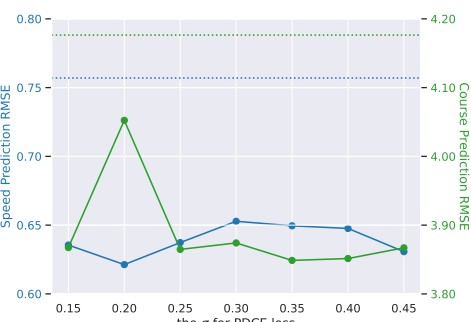

Figure 4: RMSE variation of low-level speed and course predictions with different PDCE loss $\sigma$ values on the BDD-X dataset. The dashed line represents the result of using the original CE loss.

Table 6: The impact of incorporating Environmental Predicates (EP) information for retrieval, along with the number of retrieved examples $K$ in Multimodal RAG, on high-level action and justification performance in the BDD-X dataset.

| Method | K | Action | | | | Justification | | |
|--------|---|--------|---|---|---|---------------|---|---|
| | | B4↑ | C↑ | M↑ | Acc↑ | B4↑ | C↑ | M↑ |
| Base | - | 30.8 | 221.5 | 29.2 | 61.75 | 7.8 | 85.4 | 13.2 |
| RAG w/o EP | 1 | 29.4 | 219.2 | 28.5 | 59.06 | 7.3 | 74.8 | 12.6 |
| RAG w/o EP | 2 | 29.7 | 218.6 | 28.7 | 59.91 | 7.3 | 73.7 | 12.5 |
| RAG w/ EP | 1 | 38.1 | 334.8 | 35.4 | 91.47 | 8.8 | 89.2 | 13.5 |
| RAG w/ EP | 2 | 38.2 | 334.8 | 35.3 | 91.00 | 9.4 | 95.5 | 13.9 |

**Contribution from Each Module.** The influence of incorporating each module on high-level prediction using the BDD-X dataset is shown in Table 4. The results for low-level prediction are deferred to Appendix B.1. Additionally, to better reflect the action performance improvement, we introduce a new metric, termed *high-level action predicate accuracy*, for the BDD-X dataset, which maps high-level action descriptions into one of the 16 predefined actions using GPT4o prompting and calculates accuracy accordingly. Our results indicate that: (1) the adoption of PDCE loss for low-level prediction does not negatively impact high-level prediction performance; (2) post-safety verification via MLN helps correct some unsafe actions, although the base model tends to behave conservatively; (3) multimodal RAG significantly boosts performance, with high-level action predicate accuracy improving by at least 30%. Similar observations are made in the ablation study for the DriveLM dataset, as shown in Table 5.

**PDCE Loss with Different $\sigma$ Values.** We investigate the impact of varying $\sigma$ values on low-level predictions in the BDD-X dataset, as demonstrated in Figure 4. Our findings reveal that the incorporation of PDCE loss consistently yields lower RMSEs for both speed and course predictions compared to the base one which uses the original CE loss. Moreover, performance exhibits minimal sensitivity to changes in $\sigma$, indicating stability under the PDCE loss framework.

**Case study on post-safety verification w/ MLN.** In the BDD-X dataset, the most critical traffic rule is expressed as `SolidRedLight(x)` $\implies$ `¬Accelerate(x) ∧ ¬LeftPass(x) ∧ ¬Yield(x)`, while for the DriveLM dataset, the key traffic rule is `RedYieldSign(x)` $\implies$ `¬Fast(x)`. These two rules hold the highest weights in their respective MLN. Although DriveLM contains a significant number of lane-related traffic rules, their relative importance is diminished due to the high frequency of straight-driving scenarios, which constitute 76.95% of the dataset, leaving lane-changing scenes as a minor subset. A specific instance of rejecting and correcting aggressive driving behavior using MLN is depicted in Figure 5.

**Influence of Environmental Predicates on Retrieval.** Unlike RAGDriver (Yuan et al., 2024) that unified only video and control signal information for retrieval, our approach also incorporates explicit Environmental Predicate (EP) information (e.g., presence of a stop sign) extracted from both video and control signals, as demonstrated in Section 3.2. Specifically, as shown in Table 6, removing environmental predicates from the retrieval process results in performance similar to the base model. However, including these explicit predicates significantly enhances high-level prediction performance, which indicates substantial noise in the original video and control signal data, suggesting that extracting explicit binary environmental predicates for retrieval could be highly promising.

**Multimodal RAG with Different $K$.** We explore the impact of varying top $K$ selections for BDD-X dataset in Table 6. As we can see, significant improvements in high-level action prediction are achieved even with $K = 1$, and the performance is already comparable to the $K = 2$ scenario. Furthermore, selecting a larger $K$ value enhances performance in high-level justification prediction.

## 6 LIMITATION

There are still some limitations for SafeAuto that could be addressed in future work. For example, (1) the design of the distribution $\mathcal{D}(\mu, \sigma)$ for the PDCE loss could be further optimized to enhance performance. (2) The effectiveness of the safety verification depends on the quality of predicate extraction, which may be challenging when few predicates are available in certain scenarios. (3) Additionally, exploring the multimodal RAG with larger values of $K$ in the MLLM context could improve retrieval performance but may also increase computational complexity.

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

APPENDIX

# A  DETAILS ON SAFEAUTO-REASONING

## A.1  TRAFFIC RULE MAPPING

This section outlines the methodology for extracting first-order logic formulas from the California Driver Handbook [2]. Initially, all traffic rules are transformed into a structured format using GPT4o, based on the template: 'When [conditions], you should/should not [action] (unless [conditions]).' Subsequently, GPT4o is utilized again to translate the structured traffic rules into first-order logic formulas. The complete set of prompts is provided in Table 7 and Table 8.

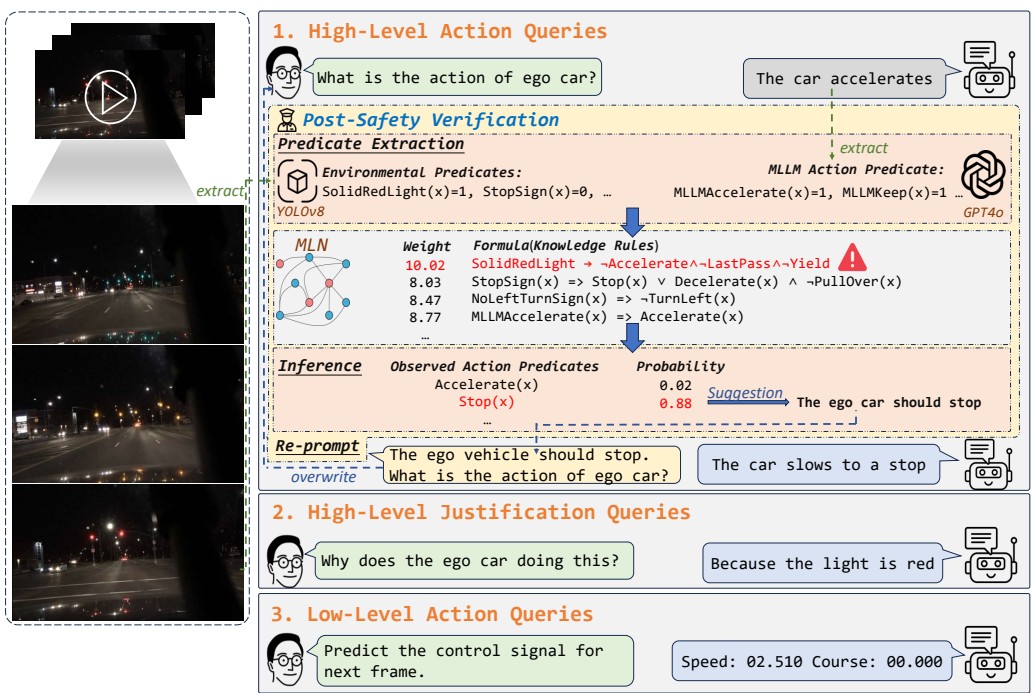

Figure 5: An example of rejecting and correcting aggressive behavior through MLN

## A.2  YOLOV8 FINE-TUNING

We fine-tuned the YOLOv8 model using the LISA dataset (Jensen et al., 2016), which contains annotations for both traffic signs and traffic signals. The dataset includes four daytime sequences and two nighttime sequences, primarily designated for testing, with a total duration of 23 minutes and 25 seconds of driving footage recorded in Pacific Beach and La Jolla, San Diego. It consists of $43,007$ frames, with annotations for $113,888$ traffic lights and $7,855$ traffic signs across $6,610$ frames. The YOLOv8m model was fine-tuned for $500$ epochs, utilizing an input image resolution of $640 \times 640$ pixels.

## A.3  PREDICATE EXTRACTION

For environmental predicates, we utilize YOLOv8 for detection, as described in Appendix A.2, for detection. To ensure consistency with RagDriver (Yuan et al., 2024), we uniformly divide video segments into 8 frames and select the final frame as the input. Additionally, in DriveLM, We leveraged the nuScenes map expansion to extract lane line information for both sides of the lane in which the ego vehicle is positioned. "For environmental predicates related to control signals in BDD-X and

---

[2] https://www.dmv.ca.gov/portal/handbook/california-driver-handbook/

As an agent for autonomous driving, your task is to extract pertinent rules from the provided text concerning autonomous driving, while simultaneously filtering out irrelevant information. In specific, please extract rules from the text relating to specific driving maneuvers listed as follows: keep, accelerate, decelerate, stop, make left turns, make right turns, reverse, merge, change lanes, park, make U-turns, overtake, yield, follow different traffic signs. Disregard unrelated actions for autonomous driving like "looking around/ checking mirrors" or similar non-quantifiable action.

Use the structured format: 'When [conditions], you should/should not [action] (unless [conditions]).' Utilize 'OR' or 'AND' to connect multiple conditions that may trigger the same action. Optionally, include 'unless [conditions]' where exceptions apply. Each rule should be direct and applicable, ensuring it aids in the precise and safe execution of self-driving maneuvers. If the text does not provide relevant advice for the actions listed, respond with 'None'.

Here is one example:

#Title#: Double Solid Yellow Lines
#Passage#: Do not pass over double solid yellow lines. Stay to the right of these lines unless you are:
 In a high-occupancy vehicle (HOV) carpool lane that has a designated entrance on the left.
 Instructed by construction or other signs to drive on the other side of the road because your side is closed or blocked.
 Turning left across a single set of double yellow lines to enter or exit a driveway or private road or make a U-turn.
Two sets of solid double yellow lines spaced two or more feet apart are considered a barrier. Do not drive on or over this barrier, make a left turn, or make a U-turn across it, except at designated openings.
#Extracted Rules#: When driving near double solid yellow lines, you should stay to the right of these lines unless: (i) You are in a high-occupancy vehicle (HOV) carpool lane that has a designated entrance on the left; (ii) You are instructed by construction or other signs to drive on the other side of the road because your side is closed or blocked; (iii) You are turning left across a single set of double yellow lines to enter or exit a driveway or private road, or to make a U-turn.
When two sets of solid double yellow lines spaced two or more feet apart are present, you should not drive on or over this barrier, make a left turn, or make a U-turn across it, unless there is a designated opening for such maneuvers.

Now, extract the rules for the following passage:
#Title#: {title}
#Passage#: {passage}
#Extracted Rules#:

Table 7: Prompt for converting traffic rules to structured format

Your goal is to transform natural language driving rules into first-order logical rules for autonomous driving systems, start by identifying the relevant actions and conditions from the text. Actions must choose from predefined predicates like Keep, Accelerate, Decelerate, Stop, MakeLeftTurn, MakeRightTurn, Reverse, Merge, ChangeToLeftLane, ChangeToRightLane, Park, MakeUTurn, LeftPass, RightPass and Yield.

First, analyze the natural driving rules to identify clear obligations (required actions) and prohibitions (banned actions), explicitly ignoring any actions described as conditional permissions ("may"). Each rule will either dictate required actions under specific conditions or explicitly ban certain actions in defined scenarios. For each rule:

Identify Required Actions (Obligations): If a rule specifies an action that must be taken under certain conditions, formulate this into a logical statement using the format "Condition  Action." This represents an obligatory action.

Identify Prohibited Actions (Bans): If a rule bans certain actions in specific circumstances, express this as a logical statement using the format "Condition  Action." This captures actions that are explicitly forbidden.

Here is one example:

#Natural Rules#: When driving near double solid yellow lines, you should stay to the right of these lines unless: (i) You are in a high-occupancy vehicle (HOV) carpool lane that has a designated entrance on the left; (ii) You are instructed by construction or other signs to drive on the other side of the road because your side is closed or blocked; (iii) You are turning left across a single set of double yellow lines to enter or exit a driveway or private road, or to make a U-turn.
When two sets of solid double yellow lines spaced two or more feet apart are present, you should not drive on or over this barrier, make a left turn, or make a U-turn across it, unless there is a designated opening for such maneuvers.
#Logical Rules#: (1) LeftSingleSetDoubleYellow  InHOVCarpoolWithLeftEntrance  Construction  ChangeToLeftLane  LeftPass
AdjacentSingleSetDoubleYellow  EnterOrExitDriveway  EnterOrExitPrivateRoad  MakeLeftTurn
(2) LeftDoubleSetsDoubleYellow  DesignatedOpeningLeftTurn  MakeLeftTurn
LeftDoubleSetsDoubleYellow  DesignatedOpeningUTurn  MakeUTurn

Now, extract the first-order logical rules for the following natural rules, and label each logical rule clearly with #Logical Rules# and include an index that corresponds to the index of the original rule as shown in the example. Besides when there are only conditioanl permissions ("may") and no clear obligations or progibitions, you can simply output None.
#Natural Rules#: {rules}

Table 8: Prompt for further converting traffic rules to first-order logic formulas

DriveLM(for exanple, `HCSKeep(x)`), we also employ GPT4o for extraction. The specific details of the prompts utilized for this extraction process are provided in Table 9 and Table 10

With respect to MLLM action predicates, since the output of MLLM consists of high-level action descriptions such as "The car is slowing down to stop, we map these to predicates represented as (`MLLMDecelerate(x)`, `MLLMStop(x)`). In the BDD-X dataset, due to the increased number and complexity of high-level action descriptions for MLLM action predicates, we employ GPT4o with specifically designed prompts to extract these predicates, with detailed prompts provided in Table 11. In DriveLM, given that the question-and-answer format comprises multiple-choice questions with fixed option descriptions, we predefine mapping rules to translate high-level action descriptions into predicates, as described in Table 12.

```
Given the current speed, curvature, acceleration, and course of the car, use one velocity predicate and one
directional predicate to best describe the behavior of the car.
The velocity predicates are: Keep, Accelerate, Decelerate, Stop, Reverse.
The directional predicates are: Straight, Left, Right.
Output the predicates directly without any additional information.
Here are some examples:
#Speed#: [7.18, 5.76, 4.45, 3.30, 2.24, 1.20, 0.36]
#Curvature#: [1.32, 0.88, 0.58, 1.85, 2.74, 1.61, 0.64]
#Acceleration#: [-1.22, -1.85, -2.39, -2.22, -2.01, -1.46, -0.87]
#Course#: [0.00, -10.03, -8.33, -3.23, -0.97, -0.32, -0.08]
#Predicate#: HCSStop, HCSLeft
#Speed#: [12.31, 9.51, 7.24, 5.38, 3.67, 2.76, 3.00]
#Curvature#: [-0.00, 0.00, 0.00, -0.05, -0.18, -0.67, -0.79]
#Acceleration#: [-1.85, -2.79, -2.73, -2.23, -1.67, -0.47, 0.71]
#Course#: [0.00, 0.00, 0.00, 0.00, -20.26, -60.78, 7.17]
#Predicate#: HCSDecelerate, HCSRight
#Speed#: [1.27, 4.18, 6.83, 8.87, 10.44, 12.22, 14.45]
#Curvature#: [0.00, 0.00, 0.00, -0.00, -0.01, -0.00, -0.00]
#Acceleration#: [2.27, 2.15, 1.81, 1.35, 1.28, 1.56, 1.45]
#Course#: [0.00, -0.09, 0.00, 0.00, 0.20, 0.00, 0.00]
#Predicate#: HCSAccelerate, HCSStraight
#Speed#: {speed}
#Curvature#: {curvature}
#Acceleration#: {acceleration}
#Course#: {course}
#Predicate:
```

Table 9: Prompt for Extracting High-level Control Signal Environmental Predicates from the BDD-X Dataset

```
Given the current speed and course of the car, use one velocity predicate and one directional predicate to
best describe the behavior of the car.
The velocity predicates are: Normal, Fast, Slow, Stop.
The directional predicates are: Straight, Left, Right.
Output the predicates directly without any additional information.
Here are some examples:
#Speed#: [(4.54, 0.0), (5.34, 0.0), (5.67, 0.0), (5.7, 0.0), (6.46, 0.0), (6.63, 0.0)]
#Course#: [(1.0, 0.0), (1.0, 0.0), (1.0, 0.0), (1.0, 0.0), (1.0, 0.0), (1.0, 0.0)]
#Predicate#: HCSFast, HCSStraight
#Speed#: [(10.01, 0.0), (9.88, 0.0), (9.52, 0.0), (9.39, 0.0), (9.15, 0.0), (8.94, 0.0)]
#Course#: [(0.84, 0.0), (0.84, 0.0), (0.86, 0.0), (0.89, 0.0), (0.93, 0.0), (0.95, 0.0)]
#Predicate#: HCSFast, HCSRight
#Speed#: [(2.51, 0.0), (2.49, 0.0), (2.45, 0.0), (2.43, 0.0), (2.43, 0.0), (2.37, 0.0)]
#Course#: [(0.85, 0.0), (0.85, 0.0), (0.86, 0.0), (0.85, 0.0), (0.82, 0.0), (0.75, 0.0)]
#Predicate#: HCSSlowly, HCSLeft
#Speed#: [(1.65, 0.0), (1.37, 0.0), (0.73, 0.0), (0.09, 0.0), (0.0, 0.0), (0.0, 0.0), (0.0, 0.0), (0.0, 0.0)]
#Course#: [(0.86, 0.0), (0.86, 0.0), (0.87, 0.0), (0.86, 0.0), (0.86, 0.0), (0.86, 0.0), (0.85, 0.0), (0.84,
0.0)]
#Predicate#: HCSStop, HCSStraight
#Speed#: {speed}
#Course#: {course}
#Predicate#:
```

Table 10: Prompt for Extracting High-level Control Signal Environmental Predicates from the DriveLM Dataset

## A.4 TRAINING DETAILS

The learning rate for the Markov Logic Network (MLN) is set at $1 \times 10^{-5}$. To mitigate the risk of overfitting and to avoid excessive reliance on frequently occurring scenarios, such as straight movements, regularization is incorporated into the training process, also set at $1 \times 10^{-5}$. The models are trained for a total of 300 epochs, unless interrupted by a predefined early stopping criterion: specifically, if the model's accuracy fails to improve by more than $1 \times 10^{-6}$ over 10 consecutive epochs, training will be terminated.

```
Given the current behavior of the car, please use predicates below to best describe the behavior of the car.
The predicates are:
Keep, Accelerate, Decelerate, Stop, Reverse, TurnLeft, TurnRight, UTurn, Merge, LeftPass, RightPass, Yield,
ChangeToLeftLane, ChangeToRightLane, Park, PullOver.
Here are some examples:
#Current Behavior#: The car is travelling down the road.
#Predicates#: Keep
#Current Behavior#: The car is making left turn.
#Predicates#: TurnLeft
#Current Behavior#: The car is slowing down and then comes to a stop.
#Predicates#: Decelerate, Stop
#Current Behavior#: The car is accelerating and then turns right.
#Predicates#: Accelerate, TurnRight
#Current Behavior#: The car is making a left turn and accelerates.
#Predicates#: TurnLeft, Accelerate
#Current Behavior#: The car decelerates and stops.
#Predicates#: Decelerate, Stop

Now the current behavior of the car is described, provide the predicates that best describe the behavior of
the car.

#Current Behavior#: {action}
#Predicates#:
```

Table 11: Prompt for Extracting Environmental Predicates from the BDD-X Dataset

| High-level Action Description | MLLM Action Predicate |
|---|---|
| Going straight | |
| Slightly steering to the left | `MLLMStraight(x)` |
| Slightly steering to the right | |
| Driving fast | |
| Driving very fast | `MLLMFast(x)` |
| Driving slowly | `MLLMSlow(x)` |
| Driving with normal speed | `MLLMNormal(x)` |
| Not moving | `MLLMStop(x)` |
| Steering to the left | `MLLMLeft(x)` |
| Steering to the right | `MLLMRight(x)` |

Table 12: Mapping of High-level Action Descriptions to MLLM Action Predicates

## A.5 POST-VERIFICATION DETAILS

As outlined in Section 3.2, during safety verification, we initiate the process by extracting observed grounded environmental predicates and MLLM action predicates using the object detector and GPT4o. If the final main action predicate output of the Markov Logic Network (MLN) conflicts with the suggested action from MLLM, we modify the high-level action query based on the output of the MLN. In the BDD-X dataset, we replace the original high-level action queries with new actions inferred from the MLN. For example, if the MLN predicts the possible world represented as "`Stop(x)` $= 1$" with the highest probability, we append the suggestion *"The ego vehicle should stop"* to the high-level action query. This approach facilitates the mapping back to the corresponding high-level action description and ensures the flow of conversation for subsequent queries.

In DriveLM, as high-level action queries are presented in a multiple-choice format, the final main action predicate output from the Markov Logic Network (MLN) may not always align directly to one of the options. In such cases, we filter the available options by the probability of possible worlds. Given that MLLM action predicates may map to multiple high-level action descriptions, it is feasible for multiple valid options to arise simultaneously. We then overwrite the high-level action queries by removing incorrect options and prompt the MLLM to regenerate an option.

## A.6 PREDICATES AND TRAFFIC RULES

This section provides a detailed overview of the specific aspects of the MLN construction for both the BDD-X and DriveLM datasets.

### A.6.1 BDDX

---

**Predicates**

- *Unobserved Predicates:*
  `Keep(x), Accelerate(x), Decelerate(x), Stop(x), Reverse(x), TurnLeft(x), TurnRight(x), UTurn(x), Merge(x), LeftPass(x), RightPass(x), Yield(x), ChangeToLeftLane(x),ChangeToRightLane(x),Park(x),PullOver(x)`
- *Observed Predicates:*
  - *MLLM Action Predicates:*
    `MLLMKeep(x), MLLMAccelerate(x), MLLMDecelerate(x), MLLMStop(x), MLLMReverse(x), MLLMTurnLeft(x), MLLMTurnRight(x), MLLMUTurn(x), MLLMMerge(x), MLLMLeftPass(x), MLLMRightPass(x), MLLMYield(x), MLLMChangeToLeftLane(x), MLLMChangeToRightLane(x), MLLMPark(x), MLLMPullOver(x)`
  - *Environmental Predicates:*
    `SolidRedLight(x), SolidYellowLight(x), YellowLeftArrowLight(x), RedLeftArrowLight(x), MergingTrafficSign(x), NoLeftTurnSign(x), NoRightTurnSign(x), PedCrossingSign(x), StopSign(x), RedYieldSign(x), SlowSign(x), SolidGreenLight(x), HCSKeep(x), HCSAccelerate(x), HCSDecelerate(x), HCSStop(x), HCSReverse(x), HCSStraight(x), HCSLeft(x),HCSRight(x)`

---

**Possible Worlds**

*(Keep), (Accelerate), (Decelerate), (Stop), (TurnLeft), (TurnRight), (UTurn), (PullOver), (Reverse), (Park), (Merge), (LeftPass), (RightPass), (ChangeToLeftLane), (ChangeToRightLane), (Yield), (ChangeToRightLane, Merge), (Accelerate, ChangeToRightLane), (Decelerate, Stop), (Keep, Stop), (Accelerate, Keep), (Merge, Stop), (Accelerate, LeftPass), (ChangeToLeftLane, Merge), (Stop, Yield), (Accelerate, TurnRight), (Decelerate, Keep), (Decelerate, PullOver), (ChangeToLeftLane, PullOver), (ChangeToRightLane, Stop), (Keep, TurnRight), (PullOver, Stop), (Park, Stop), (Decelerate, TurnRight), (Keep, LeftPass), (Accelerate, ChangeToLeftLane), (Accelerate, TurnLeft), (Accelerate, Stop), (Keep, TurnLeft), (Accelerate, Merge), (Decelerate, TurnLeft), (Park, PullOver), (Keep, Merge), (Keep, Park), (TurnLeft, TurnRight), (TurnLeft, Reverse), (TurnRight, Stop), (ChangeToLeftLane, Decelerate), (ChangeToRightLane, Decelerate), (TurnLeft, Stop), (TurnRight, Park), (ChangeToLeftLane, ChangeToRightLane), (Keep, RightPass), (ChangeToLeftLane, Stop), (Keep, PullOver), (LeftPass, RightPass), (ChangeToRightLane, Keep), (TurnRight, PullOver), (ChangeToLeftLane, Keep), (TurnRight, Reverse), (PullOver, Reverse), (ChangeToRightLane, TurnLeft), (Accelerate, Decelerate), (TurnRight, Yield), (Decelerate, Yield), (ChangeToRightLane, PullOver), (TurnLeft, PullOver), (Decelerate, TurnLeft, Stop), (Decelerate, Merge, Stop), (Decelerate, PullOver, Stop), (ChangeToRightLane, Decelerate, Stop), (ChangeToLeftLane, Decelerate, Stop), (Decelerate, TurnRight, Stop), (Accelerate, ChangeToLeftLane, ChangeToRightLane), (ChangeToRightLane, Decelerate, Merge), (ChangeToRightLane, Decelerate, Merge, Stop)*

---

**Traffic Rules**

- $SolidRedLight(x) \implies \neg Accelerate(x) \land \neg LeftPass(x) \land \neg Yield(x)$
- $SolidYellowLight(x) \implies TurnLeft(x) \lor TurnRight(x) \lor Keep(x) \lor Stop(x) \lor Decelerate \land \neg Accelerate(x)$
- $YellowLeftArrowLight(x) \implies Stop(x) \lor Decelerate(x)$
- $RedLeftArrowLight(x) \implies \neg(TurnLeft(x) \lor UTurn(x))$
- $MergingTrafficSign(x) \implies Decelerate(x)$
- $NoLeftTurnSign(x) \implies \neg TurnLeft(x)$
- $NoRightTurnSign(x) \implies \neg TurnRight(x)$
- $RedYieldSign(x) \implies Decelerate(x)$
- $SlowSign(x) \implies \neg Accelerate(x)$
- $StopSign(x) \implies Stop(x) \lor Decelerate(x) \land \neg PullOver(x)$
- $HCSKeep(x) \implies Keep(x) \lor Accelerate(x)$
- $HCSAccelerate(x) \implies Keep(x) \lor Accelerate(x)$
- $HCSDecelerate(x) \implies Decelerate(x) \lor Stop(x)$
- $HCSStop(x) \implies Decelerate(x) \lor Stop(x)$
- $HCSReverse(x) \implies Reverse(x)$
- $HCSLeft(x) \implies TurnLeft(x) \lor ChangeToLeftLane(x)$
- $HCSRight(x) \implies TurnRight(x) \lor ChangeToRightLane(x)$
- $HCSLeft(x) \land MLLMChangeToRightLane(x) \implies ChangeToLeftLane(x)$
- $HCSRight(x) \land MLLMChangeToLeftLane(x) \implies ChangeToRightLane(x)$
- $MLLMKeep(x) \implies Keep(x)$
- $MLLMAccelerate(x) \implies Accelerate(x)$
- $MLLMDecelerate(x) \implies Decelerate(x)$
- $MLLMStop(x) \implies Stop(x)$
- $MLLMReverse(x) \implies Reverse(x)$
- $MLLMTurnLeft(x) \implies TurnLeft(x)$
- $MLLMTurnRight(x) \implies TurnRight(x)$
- $MLLMUTurn(x) \implies UTurn(x)$
- $MLLMMerge(x) \implies Merge(x)$
- $MLLMLeftPass(x) \implies LeftPass(x)$
- $MLLMRightPass(x) \implies RightPass(x)$
- $MLLMYield(x) \implies Yield(x)$
- $MLLMChangeToLeftLane(x) \implies ChangeToLeftLane(x)$
- $MLLMChangeToRightLane(x) \implies ChangeToRightLane(x)$
- $MLLMPark(x) \implies Park(x)$
- $MLLMPullOver(x) \implies PullOver(x)$

### A.6.2 DRIVELM

**Predicates**

- ***Unobserved Predicates:***
  $Normal(x), Fast(x), Slow(x), Stop(x), Left(x), Right(x), Straight(x)$
- ***Observed Predicates:***
  - ***MLLM Action Predicates:***
    $MLLMNormal(x), MLLMFast(x), MLLMSlow(x), MLLMStop(x), MLLMLeft(x), MLLMRight(x), MLLMStraight(x)$
  - ***Environmental Predicates:***
    $SolidRedLight(x), SolidYellowLight(x), YellowLeftArrowLight(x), RedLeftArrowLight(x), MergingTraffic(x), NoLeftTurnSign(x), NoRightTurnSign(x), PedCrossingSign(x), StopSign(x), RedYieldSign(x), SlowSign(x), SolidGreenLight(x), DoubleDashedWhiteLineLeft(x), DoubleDashedWhiteLineRight(x), SingleSolidWhiteLineLeft(x), SingleSolidWhiteLineRight(x), DoubleSolidWhiteLineLeft(x), DoubleSolidWhiteLineRight(x), SingleZigzagWhiteLineLeft(x), SingleZigzagWhiteLineRight(x), SingleSolidYellowLineLeft(x), SingleSolidYellowLineRight(x), HCSNormal(x), HCSFast(x), HCSSlow(x), HCSStop(x), HCSLeft(x), HCSRight(x), HCSStraight(x)$

**Possible Worlds**

*(Normal, Left), (Normal, Right), (Normal, Straight), (Fast, Left), (Fast, Right), (Fast, Straight), (Slow, Left), (Slow, Right), (Slow, Straight), (Stop, Left), (Stop, Right), (Stop, Straight),*

**Traffic Rules**

- $SolidRedLight(x) \implies \neg Fast(x)$
- $SolidYellowLight(x) \implies \neg Fast(x)$
- $YellowLeftArrowLight(x) \implies Stop(x) \lor Slow(x)$
- $RedLeftArrowLight(x) \implies \neg Left(x)$
- $MergingTrafficSign(x) \implies \neg Fast(x)$
- $NoLeftTurnSign(x) \implies \neg Left(x)$
- $NoRightTurnSign(x) \implies \neg Right(x)$
- $RedYieldSign(x) \implies \neg Fast(x)$
- $SlowSign(x) \implies \neg Fast(x)$
- $SingleSolidWhiteLineLeft(x) \implies \neg Left(x)$
- $SingleSolidWhiteLineRight(x) \implies \neg Right(x)$
- $DoubleSolidWhiteLineLeft(x) \implies \neg Left(x)$
- $DoubleSolidWhiteLineRight(x) \implies \neg Right(x)$
- $SingleZigzagWhiteLineLeft(x) \implies \neg Stop(x)$
- $SingleZigzagWhiteLineRight(x) \implies \neg Stop(x)$
- $HCSNormal(x) \implies Normal(x)$
- $HCSFast(x) \implies Fast(x)$
- $HCSSlow(x) \implies Slow(x)$
- $HCSStop(x) \implies Stop(x)$
- $HCSLeft(x) \implies Left(x)$
- $HCSRight(x) \implies Right(x)$
- $HCSStraight(x) \implies Straight(x)$
- $MLLMNormal(x) \implies Normal(x)$
- $MLLMFast(x) \implies Fast(x)$
- $MLLMSlow(x) \implies Slow(x)$
- $MLLMStop(x) \implies Stop(x)$
- $MLLMLeft(x) \implies Left(x)$
- $MLLMRight(x) \implies Right(x)$
- $MLLMStraight(x) \implies Straight(x)$

# B    EXTRA ABLATION STUDY

## B.1    LOW-LEVEL PREDICTION ON BDD-X

Table 13 presents an ablation study evaluating the contribution of each module in SafeAuto to the low-level control signal prediction on the BDD-X dataset. Interestingly, we find that the MLN reasoning and RAG modules have only a minimal impact on the low-level prediction accuracy, with the primary improvement stemming from the PDCE loss, as expected. Additionally, we observe that incorporating RAG slightly increases the RMSE for speed prediction but decreases the RMSE for course prediction.

Table 13: Ablation study of the contribution from each module in SafeAuto focusing on low-level control signal assessment on the BDD-X dataset.

| Method | Speed | | | | | | Course | | | | | |
|---|---|---|---|---|---|---|---|---|---|---|---|---|
| | RMSE↓ | $A_{0.1}$↑ | $A_{0.5}$↑ | $A_{1.0}$↑ | $A_{5.0}$↑ | $A_{10.0}$↑ | RMSE↓ | $A_{0.1}$↑ | $A_{0.5}$↑ | $A_{1.0}$↑ | $A_{5.0}$↑ | $A_{10.0}$↑ |
| Base | 0.76 | 53.65 | 87.38 | 95.10 | 99.76 | 99.81 | 4.18 | 76.31 | 89.87 | **94.49** | 98.21 | 99.15 |
| PDCE | **0.63** | **55.63** | 88.04 | 95.24 | **99.86** | **99.91** | 3.89 | 76.64 | 89.97 | 94.35 | 98.21 | 99.20 |
| PDCE+MLN | 0.64 | 55.58 | 87.99 | 95.24 | 99.81 | **99.91** | 3.89 | **76.68** | **90.01** | 94.35 | 98.21 | 99.20 |
| PDCE+RAG | 0.65 | 55.49 | 88.79 | **95.34** | 99.81 | **99.91** | **3.85** | 76.31 | 89.68 | 94.07 | **98.30** | **99.25** |
| PDCE+MLN+RAG | 0.65 | 55.49 | **88.84** | **95.34** | 99.81 | **99.91** | **3.85** | 76.26 | 89.68 | 94.11 | **98.30** | **99.25** |

