# OpenReview forum: "SafeAuto: Knowledge-Enhanced Safe Autonomous Driving with Multimodal Foundation Models"
_ICLR.cc/2025/Conference — Submitted to ICLR 2025_

### Official Review · Reviewer_1Fdq · 2024-11-03

**Soundness:** 3
**Presentation:** 3
**Contribution:** 3
**Rating:** 6
**Confidence:** 4

**Summary:**

The paper presents "SafeAuto," a framework for enhancing autonomous driving through multimodal foundation models. It focuses on high-level action prediction and justification, as well as low-level motion prediction, leveraging datasets like BDD-X and DriveLM. The authors claim improvements in performance metrics such as BLEU, CIDEr, and METEOR. However, the paper largely reads like a technical report, lacking significant theoretical contributions or novel methodologies, which raises questions about its overall impact and originality.

**Strengths:**

1. The paper provides a comprehensive set of experimental results demonstrating improvements in performance metrics, which is a solid contribution.
2. The experimental design and results are well-organized and clearly presented.
3. The plug-and-play nature of the framework allows it to be integrated with existing multimodal learning methods

**Weaknesses:**

1. The paper appears to rely on established methodologies without introducing substantial theoretical advancements, limiting its originality and depth.
2. While the components such as the PDCE loss and post-safety verification are interesting, they do not significantly diverge from existing techniques.
3. There is little discussion on how different predicates are selected or their impact on the overall prediction performance
4. The evaluation does not seem to adequately address edge cases or scenarios where traditional models might fail. A deeper examination of how the framework handles uncommon but critical driving situations could provide insights into its robustness and safety.

**Questions:**

1. Can the authors elaborate on the specific mechanisms used to extract and integrate environmental predicates into the retrieval process? How do these predicates influence decision-making during high-level action predictions?
2. The paper mentions a gradual increase in the value of σ during training for the PDCE loss. How was this approach determined, and what empirical evidence supports its effectiveness compared to alternative strategies?
3. How does the framework handle potential conflicts between high-level predictions and low-level control signals?

---

> ### Author Response · Authors · 2024-11-22
> **Response to Reviewer 1Fdq (Part-1)**
>
> We are deeply grateful to the reviewer for their insightful and thorough feedback, and we appreciate the recognition of our work's contribution! The suggestions and comments made for our work have significantly helped to improve its quality.
>
> > **Q1: While the components such as the PDCE loss and post-safety verification are interesting, they do not significantly diverge from existing techniques.**
>
> Thank you for your valuable feedback! We apologize for any confusion regarding the PDCE loss and post-safety verification components.
>
> **PDCE Loss**: The standard Cross-Entropy (CE) loss evaluates predictions at the token (digit) level, which can lead to situations where numerically closer predictions do not correspond to lower loss values. For instance, predicting "11.99" for the target "12.46" may result in a higher CE loss than predicting "14.46," even though "11.99" is numerically closer to "12.46." This occurs because CE loss emphasizes the correctness of individual digits over the overall numerical proximity. Our proposed PDCE loss is the first work trying to address this limitation by bridging the gap between CE and MSE losses for numerical string predictions. PDCE ensures that numerically closer predictions correspond to lower loss values through a combination of positional weighting and a Gaussian-based probability distribution.
>
> **Post-Safety Verification**: While post-safety verification has been explored in other domains, its application in autonomous driving, particularly in conjunction with LLMs, is limited. In our work, we first emphasize the critical role of post-safety verification and introduce a concrete framework based on Markov Logic Networks. This framework enhances the safety and reliability of autonomous driving systems by systematically verifying safety constraints after the model's predictions.
>
> We will make our contributions clearer in the revision,thank you for bringing these points to our attention!
>
>
> > **Q2: There is little discussion on how different predicates are selected or their impact on the overall prediction performance**
>
> Thank you for your insightful comment! Our approach to selecting predicates begins with crawling the California Driver Book. We map all natural language descriptions into first-order logical rules, such as “red light” ⇒ “stop”. From these rules, we extract and collect all the predicates used in the formulas. We then select the predicates that can be reliably detected by tools like YOLOv8 or are already provided in the dataset, such as DoubleDashedWhiteLineLeft, SingleZigzagWhiteLineLeft, and SingleSolidYellowLineRight. Detailed information about this process is provided in Appendix A.
>
> The impact of different predicates on overall prediction performance is reflected in the weight of the formulas they participate in as shown in Figure 1. Predicates that are part of higher-weighted formulas will usually have a more significant influence on the prediction outcomes.
>
> In the BDD-X dataset, the most critical traffic rule is expressed as SolidRedLight(x) ⇒ ¬Accelerate(x)∧¬LeftPass(x)∧¬Yield(x), while for the DriveLM dataset, the key traffic rule is RedYieldSign(x) =⇒ ¬Fast(x). Thus the corresponding predicates shown in these rules are the most important.
>
> To further quantitatively illustrate this, we conducted additional experiments on the number of predicates used in the BDD-X dataset. The results are summarized in the table below:
>
> | Top-k Environmental Predicates (EP) | Accuracy (%) |
> | ----------------------------------- | ------------ |
> | Top 5 EP                            | 87.75        |
> | Top 10 EP                           | 92.10        |
> | Top 15 EP                           | 92.18        |
> | All predicates                      | 92.18        |
>
> *Top-k* refers to selecting the top *k* predicates with the highest average weights based on the formulas they are involved in. As shown, increasing the number of environmental predicates leads to improved high-level action prediction accuracy.
>
> We will include a more detailed discussion and exploration of predicate selection and its impact on prediction performance in our revision. Thank you for bringing this to our attention!

---

> > ### Author Response · Authors · 2024-11-22
> > **Response to Reviewer 1Fdq (Part-2)**
> >
> > > **Q3: The evaluation does not seem to adequately address edge cases or scenarios where traditional models might fail. A deeper examination of how the framework handles uncommon but critical driving situations could provide insights into its robustness and safety.**
> >
> > Thank you for your insightful suggestion. In our work, we focus on enhancing both high-level and low-level action predictions from MLLMs in autonomous driving by introducing the PDCE loss for improved numerical precision in low-level control and implementing post-safety verification using Markov Logic Networks to ensure that any unsafe actions generated by the MLLM are identified and corrected before execution. Additionally, our multimodal RAG approach leverages previous driving scenarios to reduce hallucinations and enhance prediction reliability.
> >
> > While we have included an example in Figure 5 of Appendix A.1 demonstrating the rejection and correction of aggressive behavior in one critical driving scenario, we acknowledge the need for a more comprehensive examination of how our framework handles uncommon but critical driving situations. Thank you for helping us improve our work.
> >
> >
> > > **Q4: Can the authors elaborate on the specific mechanisms used to extract and integrate environmental predicates into the retrieval process? How do these predicates influence decision-making during high-level action predictions?**
> >
> > Thank you for the insightful question. To extract and integrate environmental predicates into the retrieval process, we first concatenate all environmental predicates into a single binary vector. This vector is then processed by a MLP to map it into a hidden embedding that aligns with embeddings from other modalities, such as video and control signals.
> >
> > We found that incorporating these binary vectors is crucial for enhancing retrieval performance, as demonstrated by our ablation study in Table 6. The results show that the average action accuracy improves by 30% when this binary compact information is included. This significant improvement indicates that the original video embeddings contain considerable noise, and integrating environmental predicates helps achieve more accurate and reliable retrieval. By leveraging these predicates, our model can make more informed and precise high-level action predictions. We will emphasize this point more in our revision.
> >
> > > **Q5: The paper mentions a gradual increase in the value of σ during training for the PDCE loss. How was this approach determined, and what empirical evidence supports its effectiveness compared to alternative strategies?**
> >
> > Thank you for the insightful question! Initially, we used a fixed value of σ for training. Although this approach already outperformed the baseline that employs pure Cross-Entropy loss, we observed that the model struggled to learn the target distribution effectively at the beginning (since the $\sigma$ is a bit large). To address this, we experimented with gradually (exponentially) increasing σ from 0.01 to 0.35 during training, transitioning from an easier-to-learn distribution to the target distribution. This gradual increase resulted in a more stable training process and improved performance, as demonstrated by our empirical results below:
> >
> > | Method                            | RMSE on Speed | RMSE on Course |
> > | --------------------------------- | ------------- | -------------- |
> > | Base                              | 0.76          | 4.18           |
> > | Fixed $\sigma$ = 0.35                | 0.72          | 3.95           |
> > | Gradually Increased $\sigma$ from 0.01 to 0.35 | 0.64          | 3.89           |
> >
> > The results indicate that the gradual increase in $\sigma$ leads to further reductions in RMSE for both speed and course predictions compared to the fixed $\sigma$ approach. Therefore, we adopted this strategy as it proved to enhance model performance more effectively. Thank you for pointing out this, and we will add this statistic into our revision for better clarification!
> >
> > > **Q6: How does the framework handle potential conflicts between high-level predictions and low-level control signals?**
> >
> > Thank you for your valuable question. Currently, we do observe some potential conflicts between high-level predictions and low-level control signals, although these cases are relatively rare. At present, we mitigate these conflicts implicitly by introducing more relevant context through retrieval from the multimodal RAG, which helps the LLM reduce potential conflicts. Additionally, our experiments on larger internal datasets indicate that increasing the amount of training data can also gradually decrease the frequency of these conflicts. We recognize that currently we can only mitigate the issue but cannot fully solve it. Addressing these conflicts more explicitly is indeed an important direction for our future work and warrants further exploration.

---

> > > ### Comment · Reviewer_1Fdq · 2024-11-26
> > >
> > > Thanks for the response. Some of my concerns have been addressed. After reading the responses and the comments from other reviewers, I would maintain the original score.

---

> > > > ### Author Response · Authors · 2024-11-27
> > > > **Thanks for the helpful review!**
> > > >
> > > > Thank you for your positive feedback, and we are glad to see that we have addressed most of your concerns. We sincerely appreciate the time and effort you have dedicated to helping us refine and improve our work. If you have any further questions or concerns, please feel free to let us know!

---

### Official Review · Reviewer_uk4o · 2024-11-03

**Soundness:** 2
**Presentation:** 2
**Contribution:** 2
**Rating:** 3
**Confidence:** 4

**Summary:**

This paper proposes a framework to enhance multimodal large language based autonomous driving using structured and unstructured knowledge. The main contribution of the paper is proposal of a new loss function that the authors call place dependent cross entropy loss, that is supposed to enhance accuracy of text based control instructions. They also propose building a reasoning component for converting safety regulations into first order logic rules. These rules are then used in a probabilistic graphical model along with environmental features to contruct a RAG model.

**Strengths:**

This paper proposes to use a lot of new methods for enhancing autonomous driving. It addresses multiple problems that are difficult to solve and poses a challenge for using large language models for autonomous driving related tasks. Understanding of numerical information from text is a common problem that this paper proposes to solve using a new loss function. Reasoning is also a challenging task for these systems, here the paper proposes a system of using safety regulations to logic within a graphical model. The RAG model is also meant to produce better driving experience by learning from past.

**Weaknesses:**

This paper proposes a few different changes and the goal of the proposals are not always clear. If the goal is to produce a better driver, there should be comparison with other autonomous driving works in standard NuScenes like evaluation, both open and closed loop. But, driving comparison with standard models like UniAD and VAD is not performed (Table 3).

The loss PDCE formulation is unclear. Any new loss function should be defined clearly with notation. Here the description of the loss is unclear. The properties of the loss needs to be studied in a more varied settings. The authors just produce a graph for a specific setting to motivate this loss, but better experiments need to be planned to demonstrate the behavior of this loss.

The action generation step from the LLM is not grounded in any way, so it is not clear how the approach would deal with invalid responses or hallucination, where it might generate unsafe actions.

It is not discussed how this approach could be used in practice, do the authors see this running in realtime in a vehicle or is it just used for data labeling offline? We need some runtime information for questions like this. Also, not all the tasks they want to solve would make sense in either case.

**Questions:**

1. Formulate the loss function in mathematical terms
2. Perform experiments on loss behavior with error bars and more varied situations
3. Provide runtime information
4. Discuss inference strategy
5. Produce more experiments with driving performance like NuScenes setup and UniAD / VAD.

---

> ### Author Response · Authors · 2024-11-22
> **Response to Reviewer uk4o (Part-1)**
>
> Many thanks to the reviewer for the thoughtful and detailed feedback. The expertise and time invested in this work have been instrumental in enhancing its quality!
>
> > **Q1: This paper proposes a few different changes and the goal of the proposals are not always clear. If the goal is to produce a better driver, there should be comparison with other autonomous driving works in standard NuScenes like evaluation, both open and closed loop. But, driving comparison with standard models like UniAD and VAD is not performed (Table 3). Please produce more experiments with driving performance like NuScenes setup and UniAD / VAD.**
>
> Thank you for your insightful feedback! Our primary objective is to develop a better unified driver model using the MLLM, which can provide both high-level descriptions of the current driving scenario and the corresponding low-level control signals. Currently, these aspects are usually handled separately in existing methods.
>
> Additionally, we have indeed utilized a subset of the NuScenes dataset, specifically the DriveLM dataset, in our work. The original NuScenes dataset only includes low-level action information. Therefore, we choose the DriveLM, which instead, selects a representative subset from NuScenes and supplements it with corresponding high-level description annotations. As shown in Table 2, our approach not only improves high-level prediction accuracy from 61.60% to 74.60% but also reduces the Average Displacement Error (ADE) from 1.51 to 0.80, which is comparable to the 0.84 achieved by UniAD. It is important to note that UniAD, while proficient in generating low-level control signals, can not provide any high-level descriptions of driving scenarios.
>
> Regarding Table 3, which focuses on the BDD-X dataset, this dataset is commonly used as a benchmark for evaluating high-level explanations of driving scenarios. It does not include any trajectory information for the ego or surrounding agents, which is necessary for deploying models like UniAD. Therefore, in BDD-X evaluations, UniAD is usually not included as a baseline, consistent with all other SOTA methods like ADAPT, DriveGPT4, RAGDriver, tested on the BDD-X dataset.
>
> In summary, our goal is to provide a unified model that offers accurate high-level descriptions for driving scenarios and narrows the performance gap in low-level control signals compared to models like UniAD that rely solely on regression.
>
> > **Q2: The action generation step from the LLM is not grounded in any way, so it is not clear how the approach would deal with invalid responses or hallucination, where it might generate unsafe actions.**
>
> Thank you for the insightful question! We acknowledge that ungrounded action generation by the LLM can sometimes lead to invalid responses or hallucinations, potentially resulting in unsafe actions. Therefore, to address this, we introduce the SafeAuto-Reasoning in Section 3.2 of our paper. SafeAuto-Reasoning verifies each generated action against our predefined traffic rules via Markov Logic Network (MLN) to ensure compliance and safety. Specifically, after the inference of MLN, it will output the safest action based on the current driving scenario, so in fact, the action generation step is indeed grounded within our framework. For example, as shown in Figure 5 in the Appendix, the LLM initially suggests a hallucinated unsafe action, "The car accelerates." But then, after the verification by the MLN, a corrected action, "stop," is inferred, which is then mapped to the corresponding high-level description, "The car slows to a stop." We then use this new grounded high-level action to overwrite the original unsafe action of acceleration.
>
> By implementing SafeAuto-Reasoning, we ensure that all generated actions adhere to safety rules, effectively mitigating the risks associated with invalid or unsafe outputs. We believe this approach enhances both the reliability and overall safety of the system. Thank you again for highlighting this important aspect!

---

> > ### Author Response · Authors · 2024-11-22
> > **Response to Reviewer uk4o (Part-2)**
> >
> > > **Q3: The loss PDCE formulation is unclear. Any new loss function should be defined clearly with notation. Here the description of the loss is unclear. The properties of the loss needs to be studied in a more varied setting. The authors just produce a graph for a specific setting to motivate this loss, but better experiments need to be planned to demonstrate the behavior of this loss. Please Formulate the loss function in mathematical terms, and also perform experiments on loss behavior with error bars and more varied situations**
> >
> > Thank you for your valuable feedback regarding the formulation and clarity of our PDCE loss. We apologize for any confusion caused and appreciate the opportunity to clarify our approach comprehensively.
> >
> > To address the limitations of the standard Cross-Entropy loss in handling numerical values, we introduce the PDCE loss. Below is the precise mathematical formulation, which is also detailed introduced from line 204 to line 246 in paper:
> >
> > $\mathcal{L_{\text{PDCE}}} = \sum_{i=1}^{n} w_i \cdot KL(\mathcal{P}_i \parallel \mathcal{D}(\mu_i, \sigma))$
> >
> > where:
> >
> > - $n$ is the number of digits in the numerical string.
> > - $\mathcal{P}_i$ represents the predicted probability distribution for the $i^{th}$ digit.
> > - $\mathcal{D}(\mu_i, \sigma)$ is a digit-level discrete Gaussian distribution centered at the true digit $\mu_i$ with a standard deviation $\sigma$, this ensures that digits closer to $\mu_i$ have higher probabilities, mimicking the behavior of MSE loss in promoting numerical closeness.
> > - $w_i$ is the positional weight for the $i^{th}$ digit, decreasing with the digit's position (e.g., higher weights for more significant digits). These weights are derived using a Gaussian distribution to emphasize the importance of higher-order digits.
> >
> > **Intuition and Motivation**
> >
> > The standard CE loss evaluates predictions at the token (digit) level, which can lead to scenarios where numerically closer predictions do not necessarily correspond to lower loss values. For example, predicting "11.99" for the target "12.46" may incur a higher CE loss than predicting "14.46," despite "11.99" being numerically closer to "12.46." This discrepancy arises because CE loss prioritizes the correctness of individual digits over the overall numerical proximity.
> >
> > By contrast, our PDCE loss incorporates a Gaussian-based weighting mechanism that emphasizes the significance of higher-order digits and allows for a degree of flexibility in lower-order digits. This alignment with MSE loss ensures that predictions closer to the target in a numerical sense result in lower loss values.
> >
> > **Properties of PDCE Loss**
> >
> > - **Numerical Closeness:** PDCE loss ensures that numerically closer predictions yield lower loss values when we represent them as strings, akin to MSE loss.
> > - **Positional Weighting:** Higher-order digits are given more weight, reflecting their greater impact on the numerical value.
> >
> > **Experimental Validation**
> >
> > We also have conducted extensive experiments to validate the effectiveness of the PDCE loss:
> >
> > 1. **Ablation Studies:**
> >    - **PDCE vs. CE Loss:** As shown in **Table 13** in Appendix B.1, replacing the standard CE loss with PDCE resulted in a significant reduction in Root Mean Squared Error (RMSE), demonstrating improved numerical prediction accuracy.
> >
> > 2. **Hyperparameter Analysis:**
> >    - We explored different values of $\sigma$ in the PDCE loss (refer to **Fig. 4**), observing that the PDCE consistently outperformed the CE loss across various settings, further confirming its robustness.
> >
> > 3. **Behavioral Analysis:**
> >    - **Fig. 3(b)** illustrates the loss landscape post fine-tuning with PDCE, showing a bell-shaped distribution that aligns with the desired numerical proximity behavior.
> >
> > In summary, the PDCE loss effectively bridges the gap between CE and MSE losses for numerical string predictions by ensuring that numerically closer predictions correspond to lower loss values. This is achieved through a combination of positional weighting and a Gaussian-based probability distribution.
> >
> > We will enhance the clarity of our PDCE loss formulation and its properties in our revision. Should you have any further questions or require additional clarifications, please feel free to let us know!

---

> > > ### Author Response · Authors · 2024-11-22
> > > **Response to Reviewer uk4o (Part-3)**
> > >
> > > > **Q4: It is not discussed how this approach could be used in practice, do the authors see this running in realtime in a vehicle or is it just used for data labeling offline? We need some runtime information for questions like this. Also, not all the tasks they want to solve would make sense in either case. Please provide runtime information.**
> > >
> > > Thank you for your insightful question! We have indeed deployed a MLLM internally within an autonomous vehicle. This deployment required significant engineering efforts to integrate the model effectively, and the time for each invocation is extremely fast. However, due to some confidentiality policies, we are unable to share specific runtime details for this deployment.
> > >
> > > But actually, there are some publicly available references that demonstrate similar implementations. For example, the blog posts on [Wayve AI: Lingo 2 Driving with Language](https://wayve.ai/thinking/lingo-2-driving-with-language/) and [Nuro: Lambda - The Nuro Driver's Real-Time Language Reasoning Model](https://medium.com/nuro/lambda-the-nuro-drivers-real-time-language-reasoning-model-7c3567b2d7b4) all provide valuable insights into deploying MLLMs in real AD scenarios, highlighting the fast inference speeds achievable with such models. These examples illustrate that integrating MLLMs into autonomous driving systems is now indeed becoming a viable and trending approach.
> > >
> > > In addition to our internal deployment, we have also conducted offline evaluations to provide further context on the inference speeds of our approach. Using a single NVIDIA A6000 GPU, we measured the average running times as follows:
> > >
> > > - **BDD-X Dataset:**
> > >   - Full conversation with Retrieval-Augmented Generation (RAG) context (including both high-level and low-level questions): 2.06 seconds per case.
> > >   - Full Safe-Reasoning component, including the detection of environmental predicates: 0.27 seconds per case.
> > >   - **Total average time per case:** 2.33 seconds.
> > > - **DrivingLM Dataset:**
> > >   - Full conversation with RAG context (including both high-level and low-level questions): 3.09 seconds per case.
> > >   - Full Safe-Reasoning component, including the detection of environmental predicates: 0.41 seconds per case.
> > >   - **Total average time per case:** 3.50 seconds.
> > >
> > > These results demonstrate that our approach achieves fast inference times, making it suitable for both offline data labeling and potential real-time applications in autonomous vehicles.
> > >
> > > > **Q5: Discuss inference strategy**
> > >
> > > Thank you for pointing that out! Regarding our inference strategy, we use KV-caching to accelerate the MLLM’s inference during multi-turn conversations, ensuring faster and more efficient responses. Additionally, we consistently employ greedy decoding to generate the responses. We will also release our code once the paper is accepted. We apologize for not including these details earlier and appreciate your understanding.

---

> > > > ### Author Response · Authors · 2024-11-29
> > > > **Kind Reminder of Approaching Rebuttal Deadline**
> > > >
> > > > We greatly appreciate the time and effort you've invested in reviewing our work! With the rebuttal deadline nearing, we would truly value any additional feedback you could provide. We are keen to resolve any remaining issues you might identify to improve our submission further, your suggestions are crucial to refining our work!

---

> > > > > ### Comment · Reviewer_uk4o · 2024-12-02
> > > > > **Response to authors comments**
> > > > >
> > > > > I thank the authors for their explanation on my comments. However, I still do not believe that my comments have been addressed. Specifically,
> > > > >
> > > > > Regarding Q1: The original comment was about driving performance comparison using NuScenes setup. I do understand DriveLM is derived from NuScenes, however, the task of planning is different from motion prediction. The authors have pointed to results in prediction in their response, whereas my comment was about driving (planning) performance.
> > > > >
> > > > > Regarding Q2: It is not clear in their example how this post processing step helps ensuring safe decisions, as without a timely response, it is not possible to make safe decisions. Here is what the authors prescribed: LLM response in a red light => accelerate. Post process using rule: red light == stop => re-prompt LLM to get the response: 'car slows to a stop'. So, getting to a stop in a red light required two LLM calls, how do you ensure that there is enough time to actually stop?
> > > > >
> > > > > Regarding Q3: The loss formulation is still not explained or motivated in a mathematical way. There are experiments about hyper-parameter choice, however for a key contribution like proposing a new loss function it should be scientifically motivated. Also, the experiments do not have error bars, we need to understand how much of the improvement is actually due to the new loss vs noise.
> > > > >
> > > > > Regarding Q4: It is clear that the inference times the authors mentioned are not sufficient. 3.5 sec is not acceptable latency in AD.
> > > > >
> > > > > So, after reviewing the revised manuscript and going through the authors comments, I maintain my original rating as I believe this work is not ready for publication in the current form.

---

> > > > > > ### Author Response · Authors · 2024-12-03
> > > > > > **Thanks for the insightful review!**
> > > > > >
> > > > > > Thanks for your time and expertise in helping polishing our work! We apolopize for some confusion or missing some more details in our previous comments, and hope the following explanation could help solve them.
> > > > > >
> > > > > > > **Regarding Q1**
> > > > > >
> > > > > > Sorry for the confusion, for the task of planning, is that representating the task Planning (P3): possible safe actions of the ego vehicle as shown in DriveLM [1] paper? (correct us if we understand it incorrectly). We indeed have the this question in our data, but in DriveLM, the planning task is used a context for the following more important behavior and motion prediction task, so the final performance of the planning task is actually reflected in the performance of the behavior and motion prediction as shown in the Table 2 and Table 3 shown in DriveLM. Where the behavior accuracy is the high-level action which is for predicting what safe action should be done next (e.g., turn left or accelerate). Therefore, we use the same consistent setting with it, as shown in Table 2 of our paper, we improve the safe action prediction on speed (e.g., accelerate or deceleration) from 65.40% to 81.61% and improve the action prediction on steer (e.g., turn left or turn right) from 81.61% to 81.90%.
> > > > > >
> > > > > > > **Regarding Q2**
> > > > > >
> > > > > > Sorry for missing the technical details, we employ Key-Value (KV) caching in our experiments to efficiently manage the context—which includes driving videos and control signals—across multiple LLM calls. For the second LLM call, we utilize KV-caching, which leverages the computation from the first call. This approach significantly reduces the processing time of the subsequent call to only about 0.2 to 0.3 seconds on an A6000 GPU. This optimized processing time ensures that the system remains highly responsive and capable of making timely decisions, such as stopping at a red light safely and effectively.
> > > > > >
> > > > > > > **Regarding Q3: Experiments on hyper-parameter choices are presented, but the new loss function lacks scientific justification. Additionally, error bars are needed to differentiate improvements from noise.**
> > > > > >
> > > > > > We appreciate your feedback on the hyper-parameter choices in the PDCE loss and regret any confusion caused by our initial explanation. To clarify, the PDCE loss has only one single hyper-parameter, $\sigma$, as detailed in our previous formulation $\mathcal{L_{\text{PDCE}}} = \sum_{i=1}^{n} w_i \cdot KL(\mathcal{P}_i \parallel \mathcal{D}(\mu_i, \sigma))$, and the $w_i$ here is determined by the $\sigma$ as shown in Fig 2. The impact of varying $\sigma$ is thoroughly analyzed in Figure 4 of our paper, demonstrating its influence on model performance.
> > > > > >
> > > > > > Regarding your concerns about the absence of error bars and the potential influence of noise on our results, we acknowledge the importance of robust statistical validation. To address this, Figure 4 presents a comprehensive comparison across different $\sigma$ values, consistently showing that both the Speed and Course Root Mean Square Error (RMSE) remain lower than those obtained using the conventional Cross-Entropy (CE) loss. It is important to note that the benchmark figures for the original CE loss represent the best outcomes from multiple runs with CE loss, ensuring their reliability.
> > > > > >
> > > > > > We hope this explanation reassures you of the validity and robustness of our findings, affirming that the improvements attributed to the PDCE loss are not merely due to random variations but are a result of a significant enhancement in the model's predictive accuracy.
> > > > > >
> > > > > > > **Regarding Q4**
> > > > > >
> > > > > > Sorry for the confusion; the 3.5 seconds is obtained with only one A6000 GPU, while in real deployment, it would involve more engineering work from the machine learning system, which is beyond the scope of our paper. We can only say it is actually much faster than 3.5 seconds in our real deployment, but we cannot report specific figures here due to internal confidentiality policies. However, as a reference, when the model is deployed at Together AI, the inference time can also be reduced far below 3.5 seconds.
> > > > > >
> > > > > > We understand the reviewer's concerns about how we can really turn MLLM into practical use, which is indeed an insightful question worth exploring. Currently, from our perspective, we believe the advantages of MLLM are its generalizability and conditionality. Therefore, in practice, we can still use models from imitation learning for normal driving scenarios and only use MLLM to take over control when the uncertainty of the prediction is high. This approach allows us to balance inference latency and practicality more effectively. We hope our work can provide some insight to make the AD safer with the increasing deployment of MLLM.
> > > > > >
> > > > > > > **Summary**:
> > > > > >
> > > > > > Thank you very much for your insightful review and the valuable time you've invested in enhancing our paper. We truly appreciate your efforts to help us improve. If you still have any additional concerns or suggestions, please let us know!

---

### Official Review · Reviewer_jpQM · 2024-11-04

**Soundness:** 2
**Presentation:** 1
**Contribution:** 3
**Rating:** 3
**Confidence:** 3

**Summary:**

The authors propose SafeAuto, a framework that can be incorporated into multimodal large language models and that consists of 3 components: (i) a new loss function called Place-Dependent Cross-Entropy, (ii) a reasoning module based on Markov Logic Networks, (iii) a multimodal Retrieval-Augmented Generation model.
The Place-Dependent Cross-Entropy loss aims to better capture numerical proximity between numeric strings generated by the large language models indicating low-level control numerical signals (e.g., speed).
The Markov Logic Network-based module serves as a post-processor that flags the large language model's outputs which do not satisfy pre-defined knowledge rules (written in first order logic and capturing relations between predicates representing actions of the ego-vehicle---such as stop or accelerate---and objects from the environment---such as a stop sign).
The multimodal Retrieval-Augmented Generation model aims at helping the decision process by retrieving driving experiences similar to the current scenario, utilising information gathered from the image embeddings, control signals and a vector containing the values of binary predicates (that capture whether certain objects, e.g. a stop sign, are present in the given image or not).
The authors conduct an experimental analysis (on two datasets), showing (i) how these three components impact the performance when compared to state-of-the-art baselines and (ii) how background knowledge can be used to enhance the predictions of multimodal large language models.

**Strengths:**

Good referencing of prior works related to most of the involved components, including recent studies.

Additionally, the paper clarifies the relation to prior works and explains how the proposed approach differs from the previous ones.

The work is clearly motivated in the paper, the research context and goals are also clearly stated.

The method is described in a way that is easy to follow.

The experimental setting is well-explained.

**Weaknesses:**

There is no discussion on how the proposed Place-Dependent Cross-Entropy Loss behaves when there is a difference in the order of magnitude between the input and the target values. From the description of the method, it appears that the Place-Dependent Cross-Entropy Loss does not take into account the decimal point position (and, for example, this means that the target probability distributions from Figure 2 for numeric string "2.69" will be the same as for numeric string "26.9").
I would appreciate it if the authors could clarify this point and the statement in lines 247-248: "Notice that when \sigma is set to 0, the loss reduces to the original definition of joint CE loss for the entire numeric string", which again is unclear as the cross-entropy loss takes into account the decimal point position.

The method offers no guarantee that reprompting the multimodal large langue models will generate a new high-level action that does not violate the knowledge rules, since such models are data-driven.

The paper is in the context of knowledge-enhanced autonomous driving and argues that the underlying models are limited as they do not account for any available knowledge when making their decision.
Specifically, the authors capture the knowledge rules in the form of logical formulae, yet the related work section does not discuss prior works that would be relevant here (e.g., representing and integrating background knowledge into neural network models).

Improvements in clarity and readability are needed. Please see below detailed comments.

Figure 2: contains pseudocode and could be moved to supplementary material, with its contents described more concisely in natural language in the main text.

Formatting: cluttered text, very little space between captions and main text (e.g., lines 73 and 237), all affecting readability.
All abbreviations should be introduced, e.g. "QA" is used without doing so.

Many typos:
- in the abstract, "the the Place-Dependent".
- lines 92, 93: "contextssuch", "signalsto"
- lines 105, 106: "scenariowhich", "informationwe"
- line 144: "knowledgespecifically", "rulesinto"
- and other such instances of words not being separated by a space when they should be.
- a space before each citation would improve the readability (line 108 "BDD-X(Kim et al., 2018) and DriveLM(Sima et al., 2023)")
- abbreviations are introduced multiple times: e.g., "cross-entropy (CE)" (lines 53, 127, 180); "Place-Dependent Cross-Entropy (PDCE)" (lines 95, 170, 397), similarly for "Markov Logic Networks", etc.
- line 277/278: "all formula"
- the title of section 6 is inconsistent with the naming convention of the rest (i.e., "Limitation.", rather than "Limitation").

Other comments:
- in Figure 2, the naming of "cumulative_probs" (line 231) could be confusing/misleading as it suggests traditional cumulative probabilities. A clearer term, such as "weights," would align with the main text (line 238).
- repetitions: e.g., lines 83-84: "regarding high-level action prediction, a significant limitation of current methods in high-level action prediction"
- line 193: "with temperature as 1.0." it is unclear what is meant here, as the temperature was not yet introduced (only later on line 374 it is introduced).

**Questions:**

One of the main components of the proposed framework is a post-processor designed to identify predictions made by the large language model that do not adhere to the knowledge rules. It would be helpful to understand how often violations of the rules were observed in the conducted experiments.

How does the method deal with cases where the large language model keeps generating a new high-level action (after reprompting) that violates the knowledge rules?

As a suggestion (also mentioned earlier, along with detailed comments), the paper would need to be improved in terms of clarity and readability.

---

> ### Author Response · Authors · 2024-11-22
> **Response to Reviewer jpQM (Part-1)**
>
> We are deeply grateful to the reviewer for their thorough and insightful feedback. Your contribution of time and expertise has significantly enriched the development of our research!
>
> > **Q1: how the proposed Place-Dependent Cross-Entropy Loss behaves when there is a difference in the order of magnitude between the input and the target values. From the description of the method, it appears that the Place-Dependent Cross-Entropy Loss does not take into account the decimal point position (and, for example, this means that the target probability distributions from Figure 2 for numeric string "2.69" will be the same as for numeric string "26.9")**
>
> Thank you for pointing this out! As detailed in the Experimental Details paragraph in Section 4, we consistently format all numeric strings to the same number of digits during training. For example, in the BDD-X dataset, we represent numbers using five digits, formatting "2.69" as "02.690" and "26.9" as "26.900." This ensures that the Place-Dependent Cross-Entropy Loss can effectively balance the loss for each numeric string. We apologize for any confusion and will clarify this point further in the revision.
>
> > **Q2: I would appreciate it if the authors could clarify this point and the statement in lines 247-248: "Notice that when \sigma is set to 0, the loss reduces to the original definition of joint CE loss for the entire numeric string", which again is unclear as the cross-entropy loss takes into account the decimal point position.**
>
> Apologies for any confusion. As shown in the paper, the PDCE loss is defined as $\sum_{i=1}^{n} w_i \cdot \text{KL}(\mathcal{P}_i \parallel \mathcal{D}(\mu_i, \sigma))$. When $\sigma$ is set to 0, the distribution $\mathcal{D}(\mu_i, \sigma)$ simplifies to $\mathcal{D}(\mu_i, 0)$, which places a probability of 1 on the target digit $\mu_i$ (which is just the hard label for digit $\mu_i$). Consequently, all $w_i$ are also 1, according to the pseudo-code shown in Fig 2. As a result, it just reduces to computing the KL divergence between the probability distribution over possible digits for a hard assigned label $\mu_i$, which is exactly the standard cross-entropy loss. Sorry for the confusion, we will clarify this further in our revision.
>
> > **Q3: The method offers no guarantee that reprompting the multimodal large language models will generate a new high-level action that does not violate the knowledge rules, since such models are data-driven.**
>
> Thank you for the insightful question. We acknowledge the potential confusion arising from our brief mention of reprompting in the main text, with more detailed explanations deferred to Appendix A.5.
>
> To clarify, during safety verification, the Markov Logic Network (MLN) outputs a corrected action, such as 'stop.' However, 'stop' is a predicate from the MLN, not a natural high-level description. Therefore, "reprompting" in this context involves mapping the MLN-provided 'stop' back to a high-level action. For example, we prepend "The ego vehicle should stop." to the query "What is the action of the ego car?" resulting in the new prompt "The ego vehicle should stop, then what is the action of the ego car?" as demonstrated in Fig 5 on Page 13. This simple mapping effectively transforms "stop" into a natural description like "The car slows to a stop," which is then used to overwrite the original erroneous response.
>
> Thus, reprompting here refers to mapping an inferred action, such as 'stop,' into an appropriate high-level description, rather than re-asking the multimodal large language model (MLLM) the same question to obtain a different response. We apologize for any confusion and will integrate these details into the main text of the revision.

---

> > ### Author Response · Authors · 2024-11-22
> > **Response to Reviewer jpQM (Part-2)**
> >
> > > **Q4: The paper is in the context of knowledge-enhanced autonomous driving and argues that the underlying models are limited as they do not account for any available knowledge when making their decision. Specifically, the authors capture the knowledge rules in the form of logical formulae, yet the related work section does not discuss prior works that would be relevant here (e.g., representing and integrating background knowledge into neural network models).**
> >
> > Thank you for your valuable feedback! We apologize for the oversight in our related work section. Initially, our intention was to keep the discussion focused and concise, primarily centered around the discussion on the application of MLLM in autonomous driving. As a result, we inadvertently provided less emphasis on the integration of knowledge into neural network models. Realizing this might lead to confusion for some readers, we will revise the section to include more pertinent related works on this topic, below is a part of it which is related with MLN.
> >
> > “Specifically, Markov Logic Networks are commonly used to represent and integrate background knowledge into neural network models. In [1], the authors first explore the possibility of encoding background knowledge, such as the shape of a stop sign for image classification, and derive the corresponding strict robustness guarantees for the predictions. Building on this foundation, [2] scales the approach by incorporating Graph Convolutional Networks to encode the logical formulas, thereby enhancing the model's ability to capture complex relational structures. Additionally, they leverage variational inference techniques to efficiently approximate the posterior distributions, facilitating scalable and robust learning.”
> >
> > We will also include more related work on this topic like probabilistic circuits [3] in our revision and thank you again for your thorough review; your comment is greatly appreciated!
> >
> > [1] Yang, Z., Zhao, Z., Wang, B., Zhang, J., Li, L., Pei, H., ... & Li, B. (2022). Improving certified robustness via statistical learning with logical reasoning. Advances in Neural Information Processing Systems, 35, 34859-34873.
> >
> > [2] Zhang, J., Li, L., Zhang, C., & Li, B. (2023, February). CARE: Certifiably robust learning with reasoning via variational inference. In 2023 IEEE Conference on Secure and Trustworthy Machine Learning (SaTML) (pp. 554-574). IEEE.
> >
> > [3] Choi, Y., Vergari, A., & Van den Broeck, G. (2020). Probabilistic circuits: A unifying framework for tractable probabilistic models. UCLA. URL: http://starai. cs. ucla. edu/papers/ProbCirc20. pdf, 6.
> >
> > > **Q5: Improvements in clarity and readability are required. For example, in Figure 2, the term "cumulative_probs" (line 231) could be confusing as it suggests traditional cumulative probabilities, whereas "weights" might be a clearer term, aligning with the main text (line 238). Additionally, there are repetitions, such as in lines 83-84: "regarding high-level action prediction, a significant limitation of current methods in high-level action prediction." Also, in line 193: "with temperature as 1.0," it is unclear what is meant here, as the temperature concept is introduced later on line 374.**
> >
> > Thank you for your careful and attentive review; we truly value it! We apologize for the issues with readability. We just found the incorrect word spacing resulted from a failed compilation of the LaTeX symbol `---` intended for `—`. This has been corrected in our revision.
> >
> > Additionally, we acknowledge the potential confusion caused by the term "cumulative_probs" and have replaced it with "weights" in the revision to enhance clarity. We have also addressed the issue of repetition within the text.
> >
> > Regarding the term "temperature," the usage in line 193 actually refers to the standard sampling temperature for large language models, whereas in line 374, it pertains to the distillation temperature used in learning the rankings from text embeddings. Although both instances use the term "temperature," they relate to different contexts and processes. We will clarify these distinctions more thoroughly in our revision to prevent any confusion. We greatly appreciate your detailed feedback!

---

> > > ### Author Response · Authors · 2024-11-22
> > > **Response to Reviewer jpQM (Part-3)**
> > >
> > > > **Q6: It would be helpful to understand how often violations of the rules were observed in the conducted experiments.**
> > >
> > > Thank you for your valuable question. We evaluated the frequency of violations across 474 selected safety-critical scenarios in the BDD-X dataset, which represent situations involving stop signs, red lights, and similar scenarios. The observed violation rates are summarized in the table below:
> > >
> > > | Method   | Violation Rate(%) |
> > > | -------- | ----------------- |
> > > | Base     | 11.64%            |
> > > | PDCE     | 8.44%             |
> > > | RAG+PDCE | 5.90%             |
> > >
> > > As we can see, our proposed RAG will also help reduce the violation rate a bit, and we will add this statistic to our paper for further clarification. Thank you for bringing this to our attention!
> > >
> > > If you have any further questions or concerns, please feel free to let us know!

---

> > > > ### Author Response · Authors · 2024-11-29
> > > > **Kind Reminder of Approaching Rebuttal Deadline**
> > > >
> > > > Thank you very much for the time and effort you've devoted to reviewing our work! As the deadline for the rebuttal is quickly approaching, we would be grateful for any further suggestions you might have. We are eager to address any additional concerns you may have to enhance our work further. Your feedback is helpful to us in further polishing our work!

---

> > > > > ### Comment · Reviewer_jpQM · 2024-12-02
> > > > >
> > > > > I thank the authors for their response.
> > > > > I have read their answers and the revised PDF version of their paper. Below are my comments.
> > > > >
> > > > > Regarding Q1. In the response on how inputs are represented, the authors mention that they "will clarify this point further in the revision".
> > > > > Perhaps I have missed it, however, apart from lines 421-423 in the paper (which were present in the original version as well), there is no additional explanation.
> > > > >
> > > > > The description should fully explain how the inputs are represented, with examples serving as illustrations rather than substitutes for a formal explanation.
> > > > > For the BDD-X dataset, the paper specifies that numbers are formatted to five digits (e.g., that 8.1 is represented as 08.100). However, for the DriveLM dataset, the paper only mentions a "four-digit format" without even providing an example.
> > > > > Thus, the description is vague and leaves room for multiple interpretations: for instance, 8.1 could be represented as "8.100", "08.10", or even "008.1", depending on the intended measure and units.
> > > > > It is important to state any such choices to ensure clarity and reproducibility.
> > > > > Further, a choice on the format raises additional questions, such as what is the maximum number that can be represented (e.g., 99.999 using the BDD-X dataset input format, but unclear for the DriveLM dataset) and how the method handles outputs exceeding these limits.
> > > > >
> > > > > Regarding Q4. My concerns about related work on how knowledge is represented and integrated in neural network models were only partially addressed by the authors' response. Further, the revised version of the paper does not include any updates on this.
> > > > > Given the focus and title of the paper, it is essential to have a discussion of related works in this area.
> > > > >
> > > > >
> > > > > Regarding Q6. I appreciate the authors' response and additional analysis, which provides information on how often the proposed methods' outputs violate the provided knowledge compared to the baseline.
> > > > > However, again, although the authors mentioned in their answer that they "will add this statistic to [their] paper for further clarification", I could not find it in the revised PDF.
> > > > >
> > > > >
> > > > > General feedback. In general, I found that the authors' responses were not adequately reflected in the revised PDF. While the responses almost always indicated that changes will be included in the revised version, the actual revision includes only minimal updates.
> > > > > Although the authors corrected typos, little effort appears to have been made to improve the readability, clarity or coherence of the paper, with one of the main contributions (the proposed loss function) still being insufficiently formalised.
> > > > > Given these problems, I do not believe the paper is ready for publication at a conference like ICLR in its current form.

---

> > > > > > ### Author Response · Authors · 2024-12-02
> > > > > > **Thanks for the helpful review! (Part-1)**
> > > > > >
> > > > > > Thank you so much for your effort and expertise in helping us polish our work! Although the rebuttal discussion period has been extended to now, the final revision deadline remains Nov. 26, so we are unable to upload the new revision at this time. Therefore, instead, we will share the changes we have made in our final revision and hope you understand. We assure you that our discussions here fully align with the final revision.
> > > > > >
> > > > > > >  **Regarding Q1**
> > > > > >
> > > > > > Sorry for missing the clarification, here is our final revised paragraph for it:
> > > > > >
> > > > > > '''For example, in the BDD-X dataset, each number is formatted to five digits, such as representing 8.1 as "08.100" during training, whereas the DriveLM dataset uses a four-digit format like "08.10". The differing number of digits is due to the signals used in each dataset: in BDD-X, speed, acceleration, and turning angle range from –99.99 to 99.99 with two decimal precision, while course ranges from –180 to 180 and is scaled by dividing by two to fit within –99.99 to 99.99. To maintain the same precision, an additional decimal place is added, resulting in five digits. In contrast, all signals in the DriveLM dataset are within –99.99 to 99.99 with two decimal precision, allowing for a consistent four-digit representation.'''
> > > > > >
> > > > > > > **Regarding Q6**
> > > > > >
> > > > > > Thank you for your suggestions. We have added the new table to Appendix A. We initially wanted to add it immediately after your comment, as we found your review quite helpful in polishing our work, and we wanted to see if the statistic is indeed insightful enough based on your new feedback. However, the revision period had already passed. But, based on the current positive feedback from you, we assure you that the new statistical table, along with corresponding clarifications, has been added in Appendix A in our current revision.

---

> > > > > > > ### Author Response · Authors · 2024-12-02
> > > > > > > **Thanks for the helpful review! (Part-2)**
> > > > > > >
> > > > > > > > **Regarding Q4**
> > > > > > >
> > > > > > > Sorry for the confusion. The reason we did not include it in the main paper is due to the lengthy discussion on the knowledge component, which would cause our paper to exceed the page limit. If we inserted it, we would have to move other important parts to the appendix. For example, as you mentioned, Figure 2 contains pseudocode and could be moved to supplementary material. However, according to Reviewer uk4o, the pseudocode is important as it clarifies the mathematical formulas presented in our paper. But we now agree with your point that the pseudocode can indeed be moved to the appendix. Therefore, we have deferred it to Appendix C in our final revision and included the following complete discussion in the related work part:
> > > > > > >
> > > > > > > '''Recent advancements in the representation and integration of background knowledge into neural network models have focused on robust, scalable, and interpretable solutions. Markov Logic Networks (MLNs) have traditionally been pivotal in this endeavor. In their work, [1] explores encoding background knowledge, such as geometric and textural features of a stop sign for image classification, into these networks, providing a foundation for certifiable robustness in model predictions . Building on this foundation, [2] introduces Graph Convolutional Networks to encode logical formulas, enhancing the network's capacity to capture complex relational structures. They also leverage variational inference to approximate posterior distributions, facilitating scalable and robust learning . Extending these concepts, Neural Markov Logic Networks (NMLNs) [3] represent a significant step forward. These networks integrate ideas from MLNs but employ neural architectures to implicitly encode logical rules, thus avoiding the need for explicitly specified first-order logic rules. Similarly, Probabilistic Logic Neural Networks (pLogicNet) [4] combine the strengths of Markov logic and knowledge graph embeddings, utilizing a variational EM algorithm to efficiently learn and reason over knowledge graphs.
> > > > > > >
> > > > > > > Further, the work by [5] on probabilistic circuits introduces a unifying framework for tractable probabilistic models, which is integral to our understanding of efficient and scalable inference mechanisms in complex networks. Frameworks like HyperSPNs [6] further introduce compact and expressive models that integrate neural networks to generate parameters, enhancing the ability to model complex probability distributions. Additionally, Probabilistic Neural Circuits (PNCs) [7] strike a balance between the tractability of probabilistic circuits and the expressive power of neural networks, enabling efficient probabilistic inference while maintaining robustness in capturing intricate data patterns. These methodologies collectively advance the integration of structured background knowledge into neural architectures, promoting more robust and interpretable learning systems.''
> > > > > > >
> > > > > > > [1] Yang, Z., Zhao, Z., Wang, B., Zhang, J., Li, L., Pei, H., ... & Li, B. (2022). Improving certified robustness via statistical learning with logical reasoning. Advances in Neural Information Processing Systems, 35, 34859-34873.
> > > > > > >
> > > > > > > [2] Zhang, J., Li, L., Zhang, C., & Li, B. (2023, February). CARE: Certifiably robust learning with reasoning via variational inference. In 2023 IEEE Conference on Secure and Trustworthy Machine Learning (SaTML) (pp. 554-574). IEEE.
> > > > > > >
> > > > > > > [3] Marra, G., & Kuželka, O. (2021, December). Neural markov logic networks. In *Uncertainty in Artificial Intelligence* (pp. 908-917). PMLR.
> > > > > > >
> > > > > > > [4] Qu, M., & Tang, J. (2019). Probabilistic logic neural networks for reasoning. *Advances in neural information processing systems*, *32*.
> > > > > > >
> > > > > > > [5] Choi, Y., Vergari, A., & Van den Broeck, G. (2020). Probabilistic circuits: A unifying framework for tractable probabilistic models. UCLA. URL: [http://starai](http://starai/). cs. ucla. edu/papers/ProbCirc20. pdf, 6.
> > > > > > >
> > > > > > > [6] Shih, A., Sadigh, D., & Ermon, S. (2021). Hyperspns: Compact and expressive probabilistic circuits. *Advances in Neural Information Processing Systems*, *34*, 8571-8582.
> > > > > > >
> > > > > > > [7] Dos Martires, P. Z. (2024, March). Probabilistic Neural Circuits. In *Proceedings of the AAAI Conference on Artificial Intelligence* (Vol. 38, No. 15, pp. 17280-17289).
> > > > > > > '''
> > > > > > >
> > > > > > > > **Summary**
> > > > > > >
> > > > > > > Overall, we greatly appreciate the reviewer's time and effort in helping to polish our work. We have learned a lot from the reviews, which were quite helpful, and we have incorporated these updates into our new revision. We hope the reviewer finds that mapping the revisions mentioned above to the new version is straightforward and requires only minimal effort (copy and paste), while the new detailed clarification or the new experiments here actually took more time and effort. We appreciate your understanding and thanks for the insightful review once again!
> > > > > > >
> > > > > > > If you still have concerns about our new revision, please let us know; your feedback is important to us.

---

### Author Response · Authors · 2024-11-22
**Revision Summarization**

We would like to thank all the reviewers for their valuable feedback and suggestions. We are pleased that the reviewers found our paper well-written and recognized it as providing a novel and systematic approach to autonomous driving via MLLMs, featuring three innovative components: PDCE loss, post-safety verification via MLN, and the multimodal RAG for learning from similar driving experiments, all of which significantly outperform other baselines. Following the reviewers' suggestions, we have made the following main revisions to further enhance our work.

- Following **Reviewer jpQM**'s suggestion, we further clarified the re-prompting of the MLLM for mapping predicates to the corresponding high-level actions.
- We added related work that is relevant to representing and integrating background knowledge into neural network models, as suggested by **Reviewer jpQM**.
- We corrected all typos and addressed the comments pointed out by **Reviewer jpQM**.
- We further explained the intuition and formulation of our PDCE loss following guidance from **Reviewer uk4o**.
- We provided detailed running time information as suggested by **Reviewer uk4o**.
- At the suggestion of **Reviewer 1Fdq**, we conducted an additional ablation study exploring the impact of the number of selected predicates on final prediction performance.
- Following **Reviewer 1Fdq**'s feedback, we added additional empirical evidence supporting the effectiveness of a gradually increasing value $\sigma$ for better performance over other strategies.

We sincerely appreciate the reviewers for dedicating their time to review our paper and look forward to any further discussion and suggestions to help improve the quality of our work.

---

### Meta-Review · Area_Chair_m2Te · 2024-12-05

**Metareview:**

This paper develops SafeAuto, a novel framework that enhances MLLM-based autonomous driving systems by incorporating both unstructured and structured knowledge. The paper has the strengths of clear motivation and some innovation in using LLM for autonomous driving. The paper has the weaknesses of lacking detailed discussion on the proposed loss function, unclear goal, and no discussion on the approach can be used in practice.

After discussion, the final scores from the reviewers are 3, 3, 6. Despite the discussions between the authors and the reviewers, the two negative reviewers remain negative on the submission and do not believe the comments have been well addressed.

The AC has checked the submission, the reviews, the rebuttal, and the discussion, and sided with the two negative reviewers and decided this work is not ready for publication in the current form. Thus a rejection is recommended.

**Additional Comments On Reviewer Discussion:**

There are discussions between the authors and reviewers. After discussions, the two negative reviewers expressed explicitly that their concerns are not fully addressed and that they think this work is not ready for publication. Many issues need further revision and investigation. AC has checked the claims and issues and agreed with the two reviewers.

---

### Decision · Program_Chairs · 2025-01-22

Reject